# GOAL: Balance Multimodal Learning with Gradient Orthogonalization and Adaptive Leveraging

## Abstract

Multimodal learning, integrating the information from multiple sensory modalities, is naturally expected to outperform the single-modal counterparts. However, the heterogeneity of multimodal data often leads to two imbalance problems that impede unimodal representation learning prior to fusion. The first problem arises from inconsistent gradient magnitudes across modalities, and the second from opposing gradient directions in a unimodal encoder due to competing losses. While recent progress is achieved by strengthening *within-modality representations*, we identify *cross-modality compatibility* as another critical factor for effective feature fusion. Jointly considering these two factors for better fusion, we propose the **G**radient **O**rthogonalization and **A**daptive **L**everaging (**GOAL**), a parameter-free gradient modification method. Specifically, guided by the principle that imbalanced dependency on each modality follows the inverse relationship with prediction variance, the **AL** dynamically re-weights gradient magnitude by utilizing prediction entropy as a variance estimator. Furthermore, the **GO** ensures a synergistic update to obtain the compatible multimodal features through the projection of conflicting gradients. Extensive experiments across various modalities and frameworks indicate that GOAL consistently and significantly outperforms existing state-of-the-art methods, providing a plug-and-play module for multimodal optimization. Our code will be made publicly available.

## 1 Introduction

To simulate human perception of the world through various sensory modalities, such as text, vision, and audio, multimodal learning has attracted increasing attention (Baltrušaitis et al., 2018; Ngiam et al., 2011; Liang et al., 2024; Yuan et al., 2025). To better leverage feature encoders from individual modalities, a common framework in multimodal learning is the "multi-encoder" architecture with both modality-specific and modality-shared losses (see upper part of Fig. 1) (Chen et al., 2020; Du et al., 2022; Zhang et al., 2022; Wei & Hu, 2024). In this setup, each modality has its own feature encoder, and the resulting features are then fused for downstream tasks such as classification. In this architecture, it is commonly assumed that integrating information from multiple modalities should lead to better perception than single modality. However, the inherent heterogeneity of multimodal data raises significant challenges to effectively leverage their correlations and complementarities, called the imbalanced multimodal learning problem (Wang et al., 2020; Peng et al., 2022). Consequently, such a problem not only limits the potential of multimodal learning but may even cause the multimodal model to perform worse than its unimodal counterparts (Wang et al., 2020; Peng et al., 2022; Li et al., 2023; Wei & Hu, 2024).

From the view of model optimization, recent studies attribute the modality imbalance problem to two key gradient-related factors with respect to unimodal encoders. The first problem is characterized by a divergence in gradient magnitudes across different unimodal encoders (Wang et al., 2020). Consequently, encoders from different modalities exhibit divergent update rates, which may lead to a situation where one modality is dominant while another lags behind (Peng et al., 2022). The second problem arises when the gradients from two competing losses oppose each other, leading to inefficient encoder updates (Wei & Hu, 2024).

To address these two issues, existing studies have primarily concentrated on the gradient modification. One research direction aims to tackle the magnitude problem by leveraging gradient modulation techniques to adjust the learning rates across different modalities, such as Gradient-Blending (GB) (Wang et al., 2020), On-the-fly Gradient Modulation (OGM) (Peng et al., 2022), Adaptive Gradient Modulation (AGM) (Li et al., 2023), and Asymmetric Representation Learning (ARL) (Wei et al., 2025). A second line of work tackles the direction problem. Methods such as MMPareto (Wei & Hu, 2024) solve this issue through a multi-objective optimization framework, aiming to identify an optimal compromise between two competing losses.

Despite achieving promising results, these works exhibit certain drawbacks that require further attention. Through analysis in Sec. 3.2, we identify that an effective unimodal feature before fusion should possess two key properties: strong representational capacity within its own modality and high compatibility across different modalities. However, current gradient-based approaches primarily focus on enhancing within-modality representational capacity but neglect cross-modality compatibility, as shown in Fig. 2a. As a result, fusion of such features may exhibit reduced performance, sometimes even worse than single-modality baselines (Li et al., 2023; Wei & Hu, 2024).

To move forward, we propose the **G**radient **O**rthogonalization and **A**daptive **L**everaging (**GOAL**), a simple but effective parameter-free gradient modification method, jointly pursuing strong *within-modality representation* and high *cross-modality compatibility*. Generally, we separate the gradient vectors of each unimodal feature encoder into two groups: one corresponding to the modality-shared loss and the other to the modality-specific loss, where the previous group is modulated by the proposed GOAL. Specially, since the gradient variance is able to reflect the imbalanced dependency on each modality (Wei et al., 2025), the AL module employs prediction entropy as a variance estimator to adaptively re-weight gradient magnitudes for alleviating the problem of modal laziness. Moreover, by the projection of conflicting gradient vectors, the GO component is inspired by PCGrad (Yu et al., 2020), ensures that the features from different modalities are compatible for better fusion. Importantly, GOAL is plug-and-play for various "multi-encoder" multimodal learning frameworks, and we develop a Pytorch module for it.

Our contributions are as follows.

- Focusing on the imbalance issues in multimodal learning, we identify a key limitation of existing paradigms in learning unimodal encoders: they often emphasize the *within-modality representation* but overlook the *cross-modality compatibility*.
- To address this limitation, we propose GOAL, a plug-and-play parameter-free module that jointly considers the above two key abilities through gradient modification.
- Extensive experiments on diverse multimodal benchmarks indicate that GOAL significantly and consistently outperforms existing SOTA methods in efficiency and generalization. For practicality, we implement it as an easy-to-use PyTorch module.

## 2 RELATED WORK

Multimodal learning, by fusing data from different modalities, has demonstrated superior performance over unimodal methods across various applications (Ramachandram & Taylor, 2017; Guo et al., 2019; Prakash et al., 2021; Jiang et al., 2022; Driess et al., 2023; Caffagni et al., 2024). To leverage well-pretrained single-modality encoders, a common multimodal learning framework is to build a "multi-encoder" architecture, which is jointly optimized via modality-specific and modality-shared objectives (Chen et al., 2020; Du et al., 2022; Zhang et al., 2022; Wei & Hu, 2024). However, the heterogeneity of multimodal data often prevents joint training strategies from fully exploiting all modalities, known as the modality imbalance issue. In this situation, the multimodal methods fail to meet expectations and may even underperform compared to the best unimodal method (Wang et al., 2020). Motivated by multi-task learning, a variety of gradient-based strategies have been proposed to modify the original gradients, either in magnitude or direction.

### 2.1 GRADIENT MAGNITUDE MODIFICATION

Wang et al. (2020) observed that different modalities overfit and generalize at different rates, a phenomenon termed the laziness problem. To address this problem, several methods have been

developed by modifying gradient magnitudes, so as to balance the learning rates across different unimodal encoders. For instance, Gradient Blending (GB) (Wang et al., 2020) computes an optimal blending of modalities based on their overfitting behaviors; On-the-fly Gradient Modulation (OGM) (Peng et al., 2022) and Adaptive Gradient Modulation (AGM) (Li et al., 2023) dynamically adjust the gradient magnitudes of every modality by continuously monitoring the discrepancy in their contributions to the learning objective; Wei et al. (2025) proved that imbalanced dependency on each modality obeying the inverse ratio of their variances contributes to optimal performance, and proposed an Asymmetric Representation Learning (ARL) strategy to assist multimodal learning.

## 2.2 GRADIENT DIRECTION MODIFICATION

Wei & Hu (2024) identified that previous methods ignored gradient conflict between modality-specific and modality-shared objectives, potentially misleading the unimodal encoder optimization. To move forward, they analyzed Pareto integration (Sener & Koltun, 2018; Lin et al., 2019) in multimodal learning scenarios and then proposed an MMPareto algorithm. As a result, it ensures a final gradient with direction that is common to all learning objectives. Such similar observation and strategies are presented in different applications (Ma et al., 2020; Dimitriadis et al., 2023; 2024).

## 2.3 ADVANTAGES OF GOAL COMPARED WITH EXISTING WORKS

Basically, existing gradient-based methods, including magnitude and direction modification ones, emphasize on promoting the *within-modality representation* ability, while less considering the *cross-modality compatibility*. By contrast, GOAL takes both objectives into account simultaneously, realized by the proposed GO and AL components. Technically, different from gradient direction modification methods such as MMPareto (Wei & Hu, 2024), reducing conflicts (induced by competing losses) within a single modality, GO mitigates conflicts across different modalities to learn more compatible features. Moreover, beneficial from the none-parameter-free construction, GOAL can be implemented as a plug-and-play PyTorch module for easy integration.

## 3 GRADIENT ORTHOGONALIZATION AND ADAPTIVE LEVERAGING (GOAL)

To alleviate the imbalance problem efficiently and effectively, we propose a parameter-free and plug-and-play method called GOAL, composed by two main modules, GO and AL.

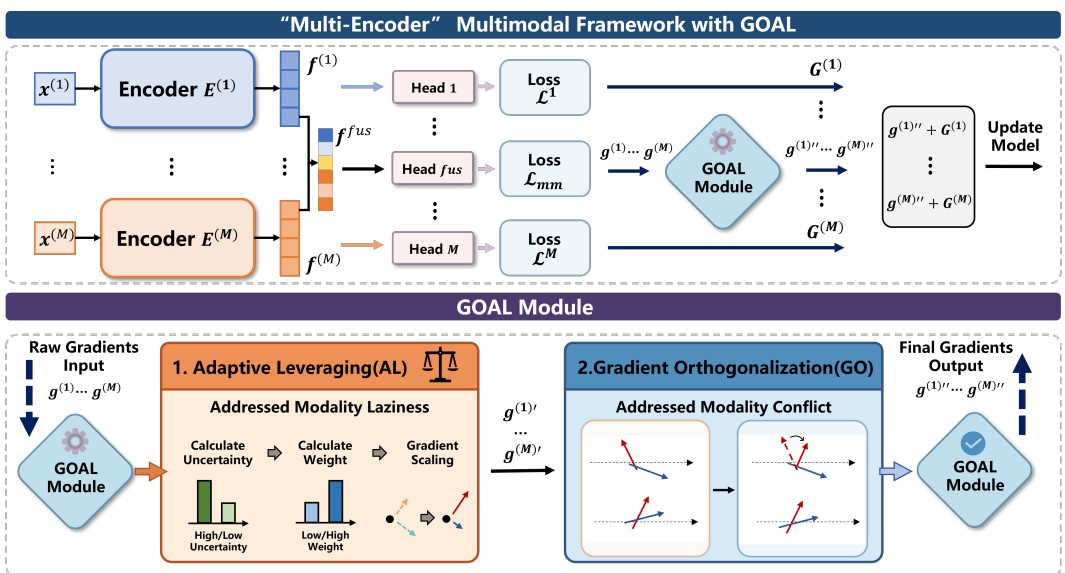

Figure 1: Illustration of "multi-encoder" multimodal framework and the implementation of GOAL, where we omit the subscript $i$ for simplicity.

### 3.1 PROBLEM FORMULATION

Let's consider a "multi-encoder" architecture involving $M$ modalities. For simple demonstration, we follow Peng et al. (2022); Li et al. (2023); Wei & Hu (2024); Wei et al. (2025) and assume a classification scenario in this section. In experiments, we will evaluate our methods on other tasks.

Specifically, a dataset composed by $N$ samples is denoted by $\mathcal{D} = \left\{ \{\boldsymbol{x}_i^{(m)}\}_{m=1}^M, y_i \right\}_{i=1}^N$, where the label $y_i \in \{1, 2, \cdots, K\}$. As shown in Fig. 1, for $i$-th sample, the $m$-th unimodal feature vector, $\boldsymbol{f}_i^{(m)} \in \mathbb{R}^{d_m}$, is obtained via the $m$-th encoder parametrized by $\{\boldsymbol{\theta}^{(m)}\}$, $i.e.$,

$$\boldsymbol{f}_i^{(m)} = E^{(m)}(\boldsymbol{x}_i^{(m)}; \boldsymbol{\theta}^{(m)}), \quad m = 1, \cdots, M. \tag{1}$$

With a fusion module $C(\cdot; \boldsymbol{\theta}^c)$, $\{\boldsymbol{f}_i^{(m)}\}_{i=1}^M$ are combined into a fused representation $\boldsymbol{f}_i^{fus} \in \mathbb{R}^{d_c}$, which is then fed into a classifier to produce the modal-fused logits $\boldsymbol{z}_i^{fus}$, formally as:

$$\boldsymbol{z}_i^{fus} = \mathbf{W}\boldsymbol{f}_i^{fus} + \boldsymbol{b}; \quad \boldsymbol{f}_i^{fus} = C(\{\boldsymbol{f}_i^{(m)}\}_{m=1}^M; \boldsymbol{\theta}^c). \tag{2}$$

To further enhance the discriminative capability of unimodal representations, existing methods also introduce a classifier for every modality to produce modal-specific logits as

$$\boldsymbol{z}_i^{(m)} = \mathbf{W}^{(m)}\boldsymbol{f}_i^{(m)} + \boldsymbol{b}^{(m)}, \quad m = 1, \cdots, M. \tag{3}$$

Based on this architecture, the total training objective $\mathcal{L}_{total}$ contains two cross-entropy (CE) losses:

$$\mathcal{L}_{total} = \mathcal{L}_{mm} + \mathcal{L}_s; \quad \mathcal{L}_{mm} = \sum_{i=1}^N \mathcal{L}_{CE}(y_i, \boldsymbol{z}_i^{fus}), \mathcal{L}_s = \sum_{m=1}^M \mathcal{L}^{(m)} = \sum_{m=1}^M \sum_{i=1}^N \mathcal{L}_{CE}(y_i, \boldsymbol{z}_i^{(m)}). \tag{4}$$

where, $\mathcal{L}_{mm}$ and $\mathcal{L}_s$ indicate the modality-fusion and the modality-specific losses, respectively.

Thus, for every unimodal encoder $E^{(m)}$, the total gradient of its parameters $\boldsymbol{\theta}^{(m)}$ is calculated by:

$$\nabla_{\boldsymbol{\theta}^{(m)}} \mathcal{L}_{total} = \nabla_{\boldsymbol{\theta}^{(m)}} \mathcal{L}_{mm} + \nabla_{\boldsymbol{\theta}^{(m)}} \mathcal{L}^{(m)} = \underbrace{\frac{\partial \mathcal{L}_{mm}}{\partial \boldsymbol{f}^{(m)}}}_{\boldsymbol{g}^{(m)}} \frac{\partial \boldsymbol{f}^{(m)}}{\partial \boldsymbol{\theta}^{(m)}} + \underbrace{\frac{\partial \mathcal{L}^{(m)}}{\partial \boldsymbol{f}^{(m)}}}_{\boldsymbol{G}^{(m)}} \frac{\partial \boldsymbol{f}^{(m)}}{\partial \boldsymbol{\theta}^{(m)}}. \tag{5}$$

To alleviate the imbalance problem across modalities, our GOAL focuses on modulating the magnitude and direction of every $\boldsymbol{g}^{(m)}$, which will be discussed below.

### 3.2 MOTIVATION OF THE GOAL

In Eq. 5, $\boldsymbol{G}^{(m)}$ originates from the modality-specific loss, enhancing within-modality representation learning by improving properties such as the discriminative power. By contrast, $\boldsymbol{g}^{(m)}$ is derived from the modality-fusion loss, serves to both enhance feature discriminability and boost cross-modality compatibility. Therefore, the compatibility among final unimodal features $\{\boldsymbol{f}^{(m)}\}_{m=1}^M$ is potentially affected if their gradient $\{\boldsymbol{g}^{(m)}\}_{m=1}^M$ exhibits conflicting directions with those of other modalities.

For better illustration, we conduct experiments based on the CREMA-D dataset, including audio and visual modalities. Specifically, Fig. 2a visualizes the histogram of the cosine similarity between $\boldsymbol{g}^{(1)}$ and $\boldsymbol{g}^{(2)}$ for a baseline method without balancing strategies, exhibiting a clear conflict between them. Moreover, in Fig. 2b, we apply Canonical Correlation Analysis (CCA) to track the evolution of the feature correlation coefficient between $\boldsymbol{f}^{(1)}$ and $\boldsymbol{f}^{(2)}$ during training. All the four methods show a consistent increase in correlation, indicating a progressive enhancement of multimodal feature compatibility. Notably, GOAL achieves the highest final coefficient, validating its greater focus on optimizing this aspect. Moreover, by incorporating our proposed GO component into existing gradient magnitude-based methods, as shown in Fig. 2c, we observe consistent performance improvements. This further demonstrates the general efficacy and motivation of our approach. The AL component is proposed to perform gradient re-weighting, maintaining consistency with prior work (Wei et al., 2025) while remaining synergistic with GO and agnostic to the fusion module.

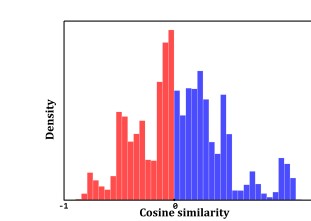 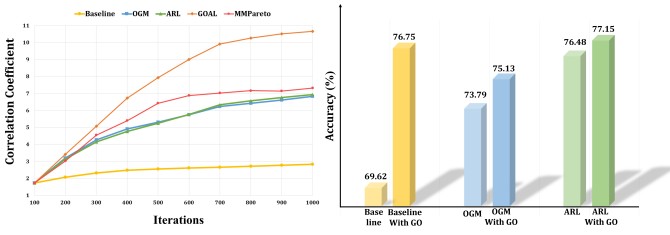

(a) Gradient conflict in baseline.  (b) Correlation coefficient.  (c) Accuracy increase with GO.

Figure 2: Motivation of the GOAL. (a) The negative cosine similarity between $g^{(1)}$ and $g^{(2)}$ for the baseline method illustrating the existence of gradient conflict between them. (b) GOAL achieves the largest feature correlation coefficient between $f^{(1)}$ and $f^{(2)}$, indicating more compatible multimodal features. (c) Incorporating GO with existing methods can further improve the performance. The experiments are conducted on the CREMA-D dataset (Cao et al., 2014).

### 3.3 ADAPTIVE LEVERAGING (AL) COMPONENT

The AL component in GOAL addresses the problem of the inconsistency in the magnitudes of the gradients $g^{(m)}$. Wei et al. (2025) theoretically demonstrated that optimal information fusion is achieved when the contribution of each modality is weighted inversely proportional to its prediction variance. Under the assumption of feature concatenation as $f_i^{fus} = [f_i^{(1)}, \cdots, f_i^{(M)}]$, Wei et al. (2025) further proposed a method to approximate their prediction variances. Building on their insight but move beyond the assumption of fusion by concatenation, we introduce a logits-based method to obtain the predictive uncertainty to re-weight the gradient magnitude.

Specifically, in classification task, the entropy of a predicted probability distribution serves as a natural measure of uncertainty and is closely correlated with the variance of the prediction (proof in Appendix A.3). In other words, given a sample $x_i^{(m)}$, if the unimodal encoder $E^{(m)}$ produces a prediction with high entropy, the prediction variance for that modality is also high. Therefore, AL modulates the magnitude of the gradient $g^{(m)}$ through the following three steps.

**Step 1: Uncertainty Calculation.** For sample $x_i^{(m)}$, we calculate its uncertainty $u_i^{(m)}$ as the normalized entropy of its prediction probability distribution $p_i^{(m)} = Softmax(z_i^{(m)})$:

$$u_i^{(m)} = \frac{H(p_i^{(m)})}{\log(K)} = -\frac{1}{\log(K)} \sum_{k=1}^{K} p_{i,k}^{(m)} \log(p_{i,k}^{(m)}), \qquad (6)$$

where the normalization factor $\log(K)$ scales $u_i^{(m)}$ to the range $[0, 1]$, thus making it independent of the number of classes.

**Step 2: Weight Calculation.** We define the prediction confidence as $c_i^{(m)} = 1 - u_i^{(m)}$, which is then used to compute the modulation weight $w^{(m)}$ for the gradient magnitude of $g^{(m)}$:

$$w^{(m)} = \frac{1}{N'} \sum_{i=1}^{N'} \frac{\exp(c_i^{(m)}/T)}{\sum_{m=1}^{2} \exp(c_i^{(m)}/T)}, \qquad (7)$$

where $N'$ is the total number of samples in one minibatch; temperature $T$ (set to 1 in all experiments) controls the sharpness of the weighting.

**Step 3: Gradient Magnitude Scaling.** The original gradient for each modality $g^{(m)}$ is scaled by its corresponding weight as:

$$g^{(m)'} = w^{(m)} \cdot g^{(m)}. \qquad (8)$$

Although the discussions above focus on classification, the proposed method can be extended to other tasks such as regression (experiment in Table 4) as long as a type of uncertainty estimation for prediction can be provided.

## 3.4 Gradient Orthogonalization (GO) Component

After re-weighting by AL, the GO component is applied to mitigate the modality conflict caused by opposing gradient directions across different $g^{(m)'}$ for higher compatibility among $\{f^{(m)}\}_{m=1}^{M}$. As shown in Figure 1 with two modalities as example, when the angle between $g^{(1)'}$ and $g^{(2)'}$ exceeds $90°$, GO removes their opposing components. Specifically, if treating $g^{(2)'}$ as the reference gradient, we compute the component of $g^{(1)'}$ that opposes the direction of $g^{(2)'}$ as $l_{g^{(1)->(2)}}$, which is then removed from $g^{(1)'}$ as:

$$g^{(1)''} = g^{(1)'} - l_{g^{(1)->(2)}}; \quad l_{g^{(1)->(2)}} = (g^{(1)'} \cdot g^{(2)'}) \frac{g^{(2)'}}{\|g^{(2)'}\|^2 + \epsilon}, \tag{9}$$

where $\epsilon$ is a small constant for numerical stability. Similar, if treating $g^{(1)'}$ as the reference gradient, we can get:

$$g^{(2)''} = g^{(2)'} - l_{g^{(2)->(1)}}; \quad l_{g^{(2)->(1)}} = (g^{(1)'} \cdot g^{(2)'}) \frac{g^{(1)'}}{\|g^{(1)'}\|^2 + \epsilon}. \tag{10}$$

In practice, the above two projections are applied iteratively and mutually for all conflicting pairs within a batch.

Finally, after AL and GO, we obtain the magnitude-direction-modified gradient $g^{(1)''}$ and $g^{(2)''}$ with both strong *within-modality representations* and high *cross-modality compatibility*. Then, any gradient-based optimized algorithm can be used to update the unimodal encoder $E^{(m)}$. GOAL is a plug-and-play module for any gradient-based optimization algorithm. In Algorithm 1, we provide the complete training loop with GOAL, where *prediction* and *loss* can be replaced according to different tasks. In Appendix 2, a Pytorch pseudo code of GOAL is provided for easy use.

---

**Algorithm 1** Optimization of "multi-encoder" architecture for multimodal classification with GOAL

1. *Input:* Get minibatch data $\mathcal{D}$.
2. *Forward:* Obtain modal-specific features $f^{(m)}$ and fused features $f^{fus}$ by Eq. 1 and 2.
3. *Prediction:* Obtain modal-specific logits $z^{(m)}$ and fused logits $z^{fus}$ by Eq. 2 and 3.
4. *Loss:* Calculate loss $\mathcal{L}_{total}$ with two components $\mathcal{L}_{mm}$ and $\mathcal{L}_{s}$ by Eq. 4
5. *Backward:* Calculate gradients by Eq. 5.
6. ***Gradient modification by GOAL***.
7. *Update:* Update all parameters by any SGD-like algorithms.

---

## 4 Experiments

The experimental evaluation is designed to assess the efficacy and generalizability of our proposed GOAL module by answering the following three questions. *i)* **Section 4.2**: Does GOAL consistently outperform existing state-of-the-art methods on a diverse range of multimodal classification benchmark as previous related works do. *ii)* **Section 4.3**: Are both the AL and GO components indispensable to performance improvement, and do they exhibit a synergistic relationship? *iii)* **Section 4.4**: How about the generalizability of GOAL across different feature encoder backbone, different multimodal tasks, and more complex data?

### 4.1 Experimental Settings

#### 4.1.1 Datasets and baselines

Following previous works, we first evaluate on three standard Audio-Visual benchmarks known for exhibiting modal imbalance. We further conduct experiments on another three Audio-Visual-Textual tri-modal datasets, and one Visual-Textual dataset to verify the generalization of our method.

**Audio-Visual Datasets.** *i)* **CREMA-D** (Cao et al., 2014) is an audio-visual emotion recognition dataset consisting of 7,442 clips with facial and vocal emotional expressions, annotated into 6 emotions (*e.g.*, happy, anger). Its key characteristic is the dominance of the audio modality, where vocal

cues often outperform visual features in predicting emotional states, leading to inherent modal imbalance. *ii)* **Kinetics-Sounds (KS)** (Arandjelovic & Zisserman, 2017) is a large-scale audio-visual video event classification dataset containing 31 human action classes for scalability testing. *iii)* **Audio-Visual Event(AVE)** (Tian et al., 2018) focuses on audio-visual event localization in unconstrained videos. It contains 4,143 10-second videos from YouTube, covering 28 event categories.

**Audio-Visual-Text Datasets.** *i)* **CMU-MOSI** (Zadeh et al., 2016) is a seminal dataset for multimodal sentiment analysis, featuring 2,199 opinion video clips from YouTube annotated for sentiment intensity. *ii)* **MELD** (Poria et al., 2018) is a large-scale, multi-party conversational emotion recognition dataset with over 13,000 utterances from the Friends TV series. It requires understanding context from three modalities within a dynamic, conversational setting, presenting a complex fusion challenge. *iii)* **IEMOCAP** (Busso et al., 2008), is a multimodal benchmark for emotion recognition, which includes interactions between ten actors. It contains approximately 12 hours of audiovisual recordings annotated with categorical emotions such as happiness, sadness, anger, and neutrality.

**Image-Text Dataset.** *i)* **Hateful Memes** (Kiela et al., 2020) is developed for multimodal hate speech detection. This dataset includes more than 10,000 multimodal samples to advance visual-linguistic classification research, particularly in identifying hateful content combining images and text. *ii)* **MM-IMDB** (Arevalo et al., 2017) is a large-scale dataset for multimodal movie analysis. It integrates visual data from movie posters and textual information from plot summaries.

**Baselines.** We compare GOAL with 8 SOTA methods: Grad-Blending (Wang et al., 2020), OGM (Peng et al., 2022), AGM (Li et al., 2023), PMR (Fan et al., 2023), MMPareto (Wei & Hu, 2024), MLA (Zhang et al., 2024), D&R (Wei et al., 2024), and ARL (Wei et al., 2025).

### 4.1.2 IMPLEMENTATION DETAILS

For fair comparison with baselines, we follow their settings. Specifically, we apply ResNet-18 (He et al., 2016) as the backbone for audio and visual modality. The audio inputs are processed in the same way as OGM (Peng et al., 2022). For video inputs, two frames of each clip are sampled for CREMA-D, CMU-MOSI, MELD and IEMOCAP datasets; three frames for Kinetics-Sounds and AVE datasets. For textual inputs, we utilize pre-trained BERT (Devlin et al., 2019) to extract embeddings. We use feature concatenation along the channel dimension for fusion, and apply the SGD optimizer (Robbins & Monro, 1951) with the momentum value 0.9 with weight decay $1 \times 10^{-4}$. The learning rate is set as $4 \times 10^{-4}$, and the classification Accuracy (Acc) as well as the macro F1-score (Sokolova & Lapalme, 2009) are used as the evaluation metric. All experiments are performed on a single NVIDIA GeForce 5090 GPU. In all tables, **bold** and underline represent the best and the second best methods, respectively.

Table 1: Performance comparison of different methods on three Audio-Visual benchmarks.

| Dataset | CREMA-D | | KS | | AVE | |
|---|---|---|---|---|---|---|
| Methods | Acc | macro F1 | Acc | macro F1 | Acc | macro F1 |
| Audio-only | 63.17 | 63.52 | 59.29 | 58.75 | 59.95 | 55.90 |
| Visual-only | 65.05 | 64.40 | 60.69 | 60.31 | 32.84 | 30.29 |
| Concatenation | 69.62 | 70.14 | 69.88 | 69.64 | 65.17 | 60.87 |
| Grad-Blending (CVPR2020) | 71.51 | 71.99 | 70.37 | 69.86 | 65.92 | 61.66 |
| OGM (CVPR2022) | 73.79 | 73.83 | 71.21 | 70.75 | 64.93 | 60.95 |
| AGM (ICCV2023) | 72.31 | 72.82 | 71.10 | 70.86 | 66.42 | 61.62 |
| PMR (CVPR2023) | 68.95 | 69.96 | 70.16 | 69.97 | 65.17 | 61.34 |
| MMPareto (ICML2024) | 74.87 | 75.57 | 72.89 | 72.36 | 67.41 | 63.29 |
| MLA (CVPR2024) | 75.54 | 76.36 | 72.36 | 71.88 | 68.66 | 63.55 |
| D&R (ECCV2024) | 75.27 | 74.85 | 71.59 | 70.77 | 68.16 | 64.92 |
| ARL (ICCV2025) | 76.48 | 76.84 | 73.06 | 72.39 | 71.02 | **68.14** |
| GOAL | **78.23** | **78.85** | **75.37** | **74.57** | **72.14** | 67.39 |

### 4.2 COMPARISON WITH SOTA METHODS ON MULTIMODAL CLASSIFICATION TASKS

We comprehensively evaluate the GOAL on three standard audio-visual benchmarks, comparing it against a naive concatenation baseline and eight SOTA methods designed for imbalanced multi-

modal learning. The results are reported in Table 1. Clearly, GOAL consistently achieves the best performance in both accuracy and macro F1 (only one second best in macro F1 on AVE). Compared to the second-best method, GOAL improves the accuracy by 1.75%, 2.31%, 1.12% on CREMA-D, KS and AVE datasets, respectively. These results demonstrate the effectiveness of GOAL for alleviating imbalanced multimodal learning problems. Further in Fig. 3, we visualize the fused features of the baseline method and our model via UMAP on the CREMA-D dataset. It can be observed that our model generates more compact and well-separated clusters than the base concatenation operation, to yield higher classification performance.

### 4.3 ABLATION STUDY FOR DIFFERENT COMPONENTS IN GOAL

To measure the contributions of GOAL's components, AL and GO, we conduct an ablation study on the CREMA-D dataset, whose results are presented in Table 2. The concatenation without any additional processing is our baseline method. Obviously, both AL and GO effectively enhance model performance compared to the baseline. The GO component, however, achieves a more substantial gain, highlighting the critical importance of resolving gradient direction conflicts. Furthermore, considering the GO component is a plug-and-play module to be incorporated into existing magnitude-oriented methods, we apply the GO to representative ARL and OGM. As shown in Fig. 2c, OGM+GO and ARL+GO perform much better than the original OGM and ARL, respectively. These results indicate that it is important to introduce a method for alleviating the gradient direction conflict between different modalities. Besides, the full GOAL model outperforms both AL-only and GO-only, demonstrating a synergistic relationship between AL and GO.

Table 2: The effects of various components of GOAL on the CREMA-D dataset.

| Method | Component | | CREMA-D | |
|--------|-----------|-----|---------|---------|
| | AL | GO | Acc | macro F1 |
| Concat | | | 69.62 | 70.14 |
| | ✓ | | 75.13 | 75.35 |
| | | ✓ | 76.75 | 77.33 |
| GOAL | ✓ | ✓ | **78.23** | **78.85** |

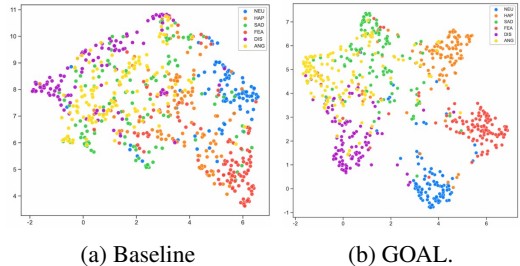

(a) Baseline       (b) GOAL.

Figure 3: Distribution of fused feature colored by different classes on the CREMA-D dataset.

### 4.4 GENERALIZATION VALIDATION

#### 4.4.1 DIFFERENT FEATURE ENCODER BACKBONES

To demonstrate that our proposed model is an architecture-agnostic optimization module, we replace the ResNet-18 with another CNN backbone, ConvNeXt (Liu et al., 2022), and also the Transformer backbone. Specifically, we apply ConvNext-B with 89M parameters and ViT-B/32 (Dosovitskiy et al., 2020) with 87M parameters on CREMA-D, Kinetics Sounds and AVE datasets.

The accuracy results are reported in Table 5, where "ConvNeXt" and "ViT" denote naive feature concatenation without any processing while "ConvNeXt†" and "ViT†" represent original methods with gradient modulation by GOAL. The results show that GOAL can also get performance improvements when applied to different baselines. GOAL's generalizability stems from its parameter-free design, which leverages the universal backpropagation process by operating directly on gradients, making it inherently generalizable across diverse deep neural network architectures.

#### 4.4.2 MORE COMPLEX DATASET IN MULTIMODAL CLASSIFICATION TASK

To demonstrate the broader applicability of GOAL beyond the above audio-visual tasks commonly addressed in prior works, we evaluate it on more complex and diverse scenarios. Specifically, we conduct experiments on two tasks: a tri-modal classification tasks including audio, vision, and text; a challenging visual-text classification task. For a fair comparison, we reproduce the baseline methods with their original settings from official public codes. Table 3 exhibits the detailed results, where

we also report the accuracy and macro F1 of different methods. Clearly, even in more complex tri-modality cases, our proposed method achieves the SOTA performance.

Table 3: Comparison of different methods on Audio-Visual-Text and Image-Text datasets. The "*" means we extend the original method to a trimodal version.

| Dataset | CMU-MOSI (*Audio/Visual/Text*) | | MELD (*Audio/Visual/Text*) | | IEMOCAP (*Audio/Visual/Text*) | | Hateful Memes (*Image/Text*) | |
|---|---|---|---|---|---|---|---|---|
| Methods | Acc | macro F1 | Acc | Macro F1 | Acc | macro F1 | Acc | macro F1 |
| Audio-only | 75.95 | 55.47 | 66.13 | 63.44 | 65.25 | 65.73 | - | - |
| Visual/Image-only | 73.47 | 50.23 | 64.95 | 62.74 | 64.33 | 65.60 | 56.50 | 45.70 |
| Text-only | 76.53 | 54.06 | 67.55 | 64.24 | 66.45 | 67.04 | 57.85 | 49.16 |
| Concatenation | 78.43 | 56.82 | 68.58 | 64.92 | 68.94 | 69.19 | 60.20 | 56.87 |
| Grad-Blending | 79.74 | 57.32 | 69.46 | 67.22 | 70.05 | 70.75 | 60.65 | 56.59 |
| OGM* | 81.05 | 57.59 | 70.34 | 67.80 | 71.43 | 71.90 | 61.30 | 58.56 |
| AGM* | 80.17 | 58.15 | 70.65 | 67.08 | 71.61 | 72.41 | 61.80 | 58.66 |
| PMR* | 79.45 | 57.53 | 68.35 | 65.82 | 69.68 | 70.41 | 60.35 | 57.25 |
| MMPareto* | 81.34 | 57.16 | 70.46 | 67.53 | 71.89 | 72.38 | 61.85 | 57.95 |
| D&R* | 80.61 | 56.63 | 71.07 | 67.93 | 70.88 | 71.64 | 61.35 | 54.60 |
| ARL* | 80.90 | 56.88 | 70.77 | 67.84 | 71.24 | 71.57 | 61.60 | 59.08 |
| GOAL | **81.78** | **58.32** | **71.23** | **68.22** | **72.90** | **73.61** | **62.30** | **60.51** |

Table 4: Performance comparison on two **regression** datasets in terms of MAE ($\downarrow$). The "*" means the version we modified for this task.

| Dataset | MM-IMDB | CMU-MOSI |
|---|---|---|
| Methods | MAE | MAE |
| Baseline | 0.7145 | 0.7708 |
| MMPareto* | 0.6978 | 0.7562 |
| ARL* | 0.7037 | 0.7521 |
| GOAL | **0.6854** | **0.7412** |

Table 5: Performance comparison using different backbones. "†" denotes the baseline backbones with GOAL.

| Dataset | CREMA-D | KS | AVE |
|---|---|---|---|
| Methods | Acc | Acc | Acc |
| ViT | 70.83 | 82.49 | 85.82 |
| ViT† | **72.45** | **83.51** | **87.06** |
| ConvNeXt | 74.46 | 84.98 | 86.32 |
| ConvNeXt† | **76.88** | **86.72** | **88.06** |

### 4.4.3 REGRESSION TASK

To demonstrate that our method is not limited to classification tasks, we validate on multimodal regression tasks. Specifically, we perform a regression task of predicting movie ratings on a image-text dataset MM-IMDB (Arevalo et al., 2017), and a regression task of predicting sentiment scores on a trimodal dataset CMU-MOSI (Zadeh et al., 2016). We adopted Mean Absolute Error (MAE) (Willmott & Matsuura, 2005) as our evaluation metric. The model was trained using Mean Squared Error loss (Ruppert, 2004) and Negative Log Likelihood Loss (Seitzer et al., 2022). More details are shown in Appendix A.4. The results are shown in Table 4. It can be seen that after applying the GOAL method, the prediction error of the model is further reduced compared to the baseline.

## 5 CONCLUSION AND FUTURE WORK

In this work, we propose to address the imbalanced multimodal learning problem via *i) within-modality representation enhancement* and *ii) cross-modality incompatibility elimination*, where recent papers primarily concentrate on the former issue. Specifically, we introduce a parameter-free gradient modification framework named GOAL in this paper, comprising the Adaptive Leveraging (AL) part and the Gradient Orthogonalization (GO) part. Among these, AL is responsible for gradient magnitude adjustment according to modality uncertainties, while GO encourages different features of different modalities to be compatible through the conflicting gradient projection. Comprehensive experiments beyond the standard audio-textual benchmarks validate the robustness and efficiency of GOAL, and we also evaluate GOAL as a plug-and-play module. In the future, GOAL is expected to support more general multimodal learning, such as "joint-encoder" architecture, model pretraining, and different tasks.

**Ethics Statement.** This work complies with the ICLR Code of Ethics. While our methods are general, they may be applied in contexts with societal implications, including risks related to bias, fairness, and privacy. We encourage responsible use and declare no conflicts of interest.

**Reproducibility Statement.** We provide detailed descriptions of our methodology, datasets, model configurations, and evaluation metrics in the main text and Appendix. Upon acceptance, we will release source code and scripts to enable full reproduction of our experiments.

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

# A APPENDIX

## A.1 PSEUDOCODE OF GOAL IN A PYTORCH-LIKE STYLE

---
**Algorithm 2** Pseudocode of GOAL in a PyTorch-like style

---

```python
def GOAL(g, z, T=1.0, eps=1e-8):
    # z: modality-specific logits (B x M x K)
    # B: batch size
    # M: number of modalities
    # K: number of classes

    p = softmax(z, dim=2)  # (B x M x K)
    log_p = log(p)  # (B x M x K)
    H = -sum(p * log_p, dim=2)  # (B x M)
    u = H / log(K)  # (B x M)
    c = 1 - u  # (B x M)
    exp_c = exp(c / T)  # (B x M)
    denom = sum(exp_c, dim=1, keepdim=True)  # (B x 1)
    weights_per_sample = exp_c / denom  # (B x M)
    w = mean(weights_per_sample, dim=0)  # (M)

    leveraged_gradients = [w[m] * g[m] for m in range(M)]

    final_gradients = leveraged_gradients.clone()

    for i in range(M):
        for j in range(M):
            if j == i:
                continue
            dot = final_gradients[i] @ leveraged_gradients[j]
            if dot < 0:
                norm_sq = (leveraged_gradients[j] @ leveraged_gradients[j]) + eps
                proj = (dot / norm_sq) * leveraged_gradients[j]
                final_gradients[i] = final_gradients[i] - proj

    return final_gradients
```

---

## A.2 TRAINING PIPELINE

---
**Algorithm 3** Pseudocode of Training Loop with GOAL in a PyTorch-like style

---

```python
for epoch in range(total_epochs):
    for minibatch_x, minibatch_y in dataloader:
        f, f_fus = Encoder_and_Fusion(minibatch_x)
        z, z_fus = classifier(f, f_fus)
        L_total = loss(minibatch_y, z, z_fus)
        g, G = L_total.backward()
        g_new = GOAL(g, z_fus)
        Theta = SGD_like_algorithm(g_new, G, Theta)
```

---

## A.3 PROOF OF THE EQUIVALENCE OF SHANNON ENTROPY AND TOTAL VARIANCE AS UNCERTAINTY PROXIES

In this section, we provide a rigorous mathematical justification for using Shannon entropy as a proxy for the prediction variance of a categorical output distribution. We first establish a formal definition for categorical variance and then demonstrate its local and global equivalence with Shannon entropy.

For a $K$-class classification task, the model output is a probability vector $\boldsymbol{p} = (p_1, \ldots, p_K)$ defining a categorical distribution. To define variance, we use the one-hot vector representation, where a prediction of class $k$ corresponds to the basis vector $\boldsymbol{e}_k \in \mathbb{R}^K$. The expectation of this random vector is $\mathbb{E}[Z] = \boldsymbol{p}$. The variance is captured by the $K \times K$ covariance matrix $\Sigma = \mathbb{E}[(Z - \mathbb{E}[Z])(Z - \mathbb{E}[Z])^T]$, which can be shown to be $\Sigma = \mathrm{diag}(\boldsymbol{p}) - \boldsymbol{p}\boldsymbol{p}^T$.

To obtain a scalar measure of dispersion, we use the Total Variance, defined as the trace of the covariance matrix, $\mathrm{Var}_T(\boldsymbol{p}) = \mathrm{Tr}(\Sigma)$.

$$\mathrm{Var}_T(\boldsymbol{p}) = \sum_{k=1}^{K} \Sigma_{kk} = \sum_{k=1}^{K} p_k(1 - p_k) = 1 - \sum_{k=1}^{K} p_k^2$$

This quantity is equivalent to the Gini impurity and represents the probability that two independent samples from the distribution $\boldsymbol{p}$ belong to different categories. It provides a principled, scalar measure of prediction variance.

We demonstrate a local equivalence between Shannon entropy, $H(\boldsymbol{p}) = -\sum p_k \log p_k$, and total variance, $\mathrm{Var}_T(\boldsymbol{p})$, by analyzing their behavior around the uniform distribution $\boldsymbol{u} = (1/K, \ldots, 1/K)$, which is the point of maximum uncertainty.

A second-order Taylor expansion of $H(\boldsymbol{p})$ around $\boldsymbol{u}$ yields:

$$H(\boldsymbol{p}) \approx H(\boldsymbol{u}) + (\boldsymbol{p} - \boldsymbol{u})^T \nabla H(\boldsymbol{u}) + \frac{1}{2}(\boldsymbol{p} - \boldsymbol{u})^T \nabla^2 H(\boldsymbol{u})(\boldsymbol{p} - \boldsymbol{u})$$

Evaluating the terms gives $H(\boldsymbol{u}) = \log K$, $\nabla H(\boldsymbol{u}) = (\log K - 1) \cdot \boldsymbol{1}$, and $\nabla^2 H(\boldsymbol{u}) = -K \cdot I$. The first-order term vanishes because $\sum(p_k - u_k) = 0$. The approximation simplifies to:

$$H(\boldsymbol{p}) \approx \log K - \frac{K}{2}\|\boldsymbol{p} - \boldsymbol{u}\|_2^2 \implies \log K - H(\boldsymbol{p}) \approx \frac{K}{2}\|\boldsymbol{p} - \boldsymbol{u}\|_2^2$$

This shows the reduction in entropy from its maximum is locally proportional to the squared Euclidean distance from the uniform distribution.

By writing $\boldsymbol{p} = \boldsymbol{u} + (\boldsymbol{p} - \boldsymbol{u})$, we find that $\sum p_k^2 = \sum(u_k + (p_k - u_k))^2 = \sum u_k^2 + \|\boldsymbol{p} - \boldsymbol{u}\|_2^2 = 1/K + \|\boldsymbol{p} - \boldsymbol{u}\|_2^2$. Therefore:

$$\mathrm{Var}_T(\boldsymbol{u}) - \mathrm{Var}_T(\boldsymbol{p}) = \left(\frac{1}{K} + \|\boldsymbol{p} - \boldsymbol{u}\|_2^2\right) - \frac{1}{K} = \|\boldsymbol{p} - \boldsymbol{u}\|_2^2$$

Combining these results, we establish a direct, local linear relationship:

$$\log K - H(\boldsymbol{p}) \approx \frac{K}{2}\left(\mathrm{Var}_T(\boldsymbol{u}) - \mathrm{Var}_T(\boldsymbol{p})\right)$$

This proves that for distributions near uniform, entropy and total variance are tightly coupled, justifying the use of one as a proxy for the other in high-uncertainty regimes.

The relationship holds globally due to a shared fundamental property. A function $f(\boldsymbol{p})$ is Schur-concave if for any two probability vectors $\boldsymbol{p}, \boldsymbol{q}$ where $\boldsymbol{q}$ is "more uniform" than $\boldsymbol{p}$ (i.e., $\boldsymbol{p}$ majorizes $\boldsymbol{q}$), it holds that $f(\boldsymbol{p}) \leq f(\boldsymbol{q})$. This property formalizes the notion of an uncertainty measure.

It is a well-known result that Shannon entropy, $H(\boldsymbol{p})$, is Schur-concave. We can also show that Total Variance, $\mathrm{Var}_T(\boldsymbol{p}) = 1 - \sum p_k^2$, is Schur-concave. The function $\phi(x) = x^2$ is convex, which implies that $g(\boldsymbol{p}) = \sum p_k^2$ is Schur-convex. Since $\mathrm{Var}_T(\boldsymbol{p}) = 1 - g(\boldsymbol{p})$, it follows that $\mathrm{Var}_T(\boldsymbol{p})$ is Schur-concave.

Because both measures are Schur-concave, they are guaranteed to respond monotonically in the same direction to any change in the uniformity of the prediction distribution. This shared symmetry provides a robust global justification for the proxy relationship, confirming that they are intrinsically linked measures of categorical dispersion.

## A.4   DETAILS OF THE REGRESSION TASKS

The MM-IMDb dataset (Arevalo et al., 2017) is designed to promote the development of models that combine multimodal data types for analysis. It primarily contains data from two modalities: Textual data: plot summaries and synopses of movies. Image information: movie posters. The metadata of this dataset contains the category of each movie data and movie rating data (such as 1-10 ratings from IMDb users) used for regression analysis as shown in Figure 4.

```
{
    "rating": 6.2,
    "distributors": [
        {
            "name": "Edison Manufacturing Company",
            "long imdb name": "Edison Manufacturing Company"
        },
        {
            "name": "Continental Commerce Company",
            "long imdb name": "Continental Commerce Company"
        }
    ],
    "runtimes": [
        "1"
    ],
    "year": 1893,
    "color info": [
        "Black and White"
    ],
    "plot": [
        "A stationary camera looks at a large anvil with a blacksmith behind it and one on either side. The smith in the middle draws a heated metal rod from the
        "Three men hammer on an anvil and pass a bottle of beer around."
    ],
    "votes": 1335,
    "title": "Blacksmith Scene",
    "smart canonical title": "Blacksmith Scene",
    "long imdb canonical title": "Blacksmith Scene (1893)",
    "certificates": [
        "USA:Unrated"
    ],
    "long imdb title": "Blacksmith Scene (1893)",
    "country codes": [
        "us"
    ],
    "smart long imdb canonical title": "Blacksmith Scene (1893)",
    "cover url": "http://ia.media-imdb.com/images/M/MV5BNDgOZDgOYWYtYzMwYiOOZjVlLWI5YzUtNzBkNjlhZWM5ODk5XkEyXkFqcGdeQXVyNDkOMDg4NDk@._V1._SX100_SY75_.jpg",
    "sound mix": [
        "lent"
    ],
    "genres": [
        "Short"
    ],
```

Figure 4: Details of MM-IMDB

The CMU-MOSI dataset (Zadeh et al., 2016) is a landmark benchmark in the fields of sentiment analysis and multimodal machine learning. It focuses on in-depth sentiment analysis of people's expressed opinions across three dimensions: visual, auditory, and linguistic.

The core of the CMU-MOSI dataset is to analyze the emotional intensity of people's opinions or comments. It quantifies this intensity. The core label of CMU-MOSI is a continuous value representing the emotional intensity of the opinion expressed in each video clip. This label is derived from the combined scoring of multiple human annotators using a professional scale. The score ranges from -3 (very negative) to +3 (very positive).

## A.5   LLM USAGE

We use LLMs to assist in the polishing of a part of the paper.

## A.6   COMPARISON OF THE COSINE SIMILARITY

To prove our GOAL can effectively mitigates the modality conflict, we visualize the cosine similarity between $g^{(1)}$ and $g^{(2)}$ on the CREMA-D dataset. Specifically, Fig. 5 visualizes the histogram of the cosine similarity between $g^{(1)}$ and $g^{(2)}$ for a baseline method without balancing strategies. We also visualizes the gradient cosine similarity after using our GOAL method.

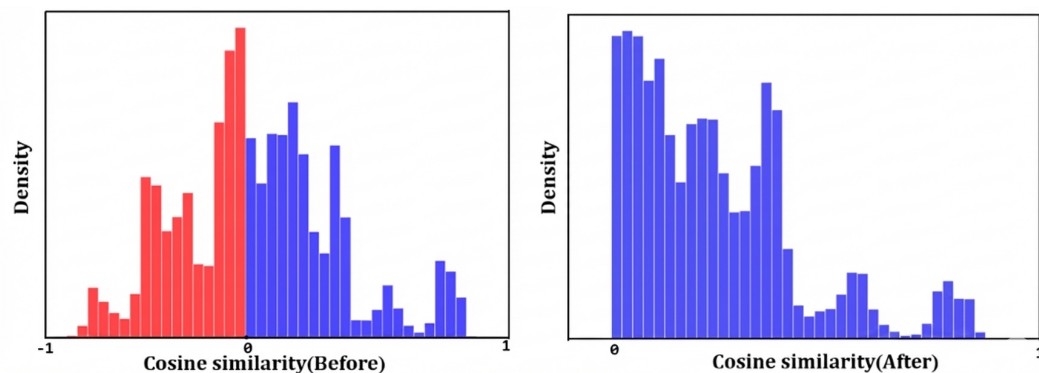

Figure 5: A comparison of the cosine similarity between $\boldsymbol{g}^{(1)}$ and $\boldsymbol{g}^{(2)}$ before and after GOAL.

## A.7 THEORETICAL PROOF FOR THE AL MODULE

The derivation reference Wei et al. (2025) here.

We consider two input modalities as $m_0$ and $m_1$. The dataset is denoted as $\mathcal{D} = \{x_i^{m_0}, x_i^{m_1}, y_i\}_{i=1,2,\ldots,N}$, where $y \in \{1, 2, \ldots, M\}$, and $M$ is the number of categories. We use two encoders $\varphi_0(\theta_0, \cdot)$ and $\varphi_1(\theta_1, \cdot)$ to extract features . The representation outputs of encoders are denoted as $z_0 = \phi_0(\theta_0, x_i^{m_0})$ and $z_1 = \phi_1(\theta_1, x_i^{m_1})$. Here, let $\phi_f(\theta_f, \cdot)$ denotes the fusion module. $W \in \mathbb{R}^{M \times (d_0 + d_1)}$ and $b \in \mathbb{R}^M$ denote the parameters of the linear classifier. The output of input $x_i$ in a multimodal model can be expressed as follows,

$$\begin{cases} f(x_i) = W z_f + b \\ z_f = \phi_f(\theta_f, z_0; z_1) \end{cases}$$

Take the most widely used vanilla fusion method, concatenation as an example, $z_f = [z_0; z_1]$ and thus $f(x_i)$ can be rewritten as follows,

$$f(x_i) = W_0 \cdot z_0 + b_0 + W_1 \cdot z_1 + b_1$$

Let $s_i^{m_0} = W_0 \cdot z_0 + b_0$, which denotes the logit output of modality $m_0$, and $s_i^{m_1} = W_1 \cdot z_1 + b_1$, which denotes the logit output of modality $m_1$. As a result, the final output is the summation of $s_i^{m_0}$ and $s_i^{m_1}$. Thus, the gradient is determined by $s_i^{m_0}$ and $s_i^{m_1}$ and the optimization dependency coefficient $d_i$ is defined as follows,

$$d = \frac{d^{m_0}}{d^{m_1}} = \frac{exp(s_i^{m_0}(y_i))/\sum_{j=1}^M exp(s_i^{m_0}(j))}{exp(s_i^{m_1}(y_i))/\sum_{j=1}^M exp(s_i^{m_1}(j))}$$

Previous work has focused on making $d$ equal to 1, but in [1] it was found that increasing the gradient of the dominant mode by a factor of 5 actually improves performance. Therefore, the best optimization dependency coefficient is not 1. And it should satisfy the following formula,

$$\begin{cases} \min_{w_0, w_1} \ g = \frac{1}{N} \sum_{i=1}^{i=N} L(f(x_i), y_i) \\ f(x_i) = w_0 s_i^{m_0} + w_1 s_i^{m_1} \end{cases}$$

where $w_0 > 0$ and $w_1 > 0$ denotes the contribution of modality $m_0$ and $m_1$, respectively, and $w_0 + w_1 = 1$. $L(.)$ measures the generalization error from prediction and groundtruth. For simplification, we define $E[f(x)] = \frac{1}{N} \sum_{i=1}^{i=N} f(x_i)$. According to the bias-variance decomposition, the error between prediction and groundtruth can be rewritten as follows,

$$\begin{cases} g = (Bias(f(x), y))^2 + Var(f(x)) + Var(\epsilon) \\ Bias(f(x), y) = E\left[f(x) - y\right] \\ Var(f(x)) = E\left[f(x)^2\right] - E\left[f(x)\right]^2 \end{cases}$$

where $Bias(f(x), y)$ is the bias of prediction, which measures the error of the prediction. $Var(f(x))$ is the variance of $f(x)$, which measures the uncertainty of the prediction, $Var(\epsilon)$ is the irreducible error in the dataset and cannot be reduced by any model. We need to minimize $Bias(.)^2$ is to minimize $Bias(s^{m_0}, y)$ and $Bias(s^{m_1}, y)$ and $Var(f(x))$.

For the term of $Var(f(x))$, we can convert it as follows,

$$Var(f(x)) = w_0^2 Var(s^{m_0}) + w_1^2 Var(s^{m_1})$$

Then, with the constraint $w_0 + w_1 = 1$, we get the solution of $w_0$ and $w_1$ as follows,

$$\begin{cases} w_0 = \frac{Var(s^{m_1})}{Var(s^{m_1}) + Var(s^{m_0})} \\ w_1 = \frac{Var(s^{m_0})}{Var(s^{m_1}) + Var(s^{m_0})} \end{cases}$$

Here, we can get the relationship between $\frac{w_0}{w_1}$ and $Var(s^{m_0})$ as well as $Var(s^{m_1})$ as follows,

$$\frac{w_0}{w_1} = \frac{\frac{1}{Var(s^{m_0})}}{\frac{1}{Var(s^{m_1})}}$$

In other words, to minimize the $Var(f(x))$ of multimodal models, we should enable the modality with a lower variance to have a larger contribution to model optimization.

A.8   ADDITIONAL INFORMATION FOR TABLE 3

Table 6: Additional information for Table 3. The "*" means we extend the original method to a trimodal version.

| Dataset | CMU-MOSI (*Audio/Visual/Text*) | | MELD (*Audio/Visual/Text*) | | IEMOCAP (*Audio/Visual/Text*) | | Hateful Memes (*Image/Text*) | |
|---|---|---|---|---|---|---|---|---|
| Methods | Acc | macro F1 | Acc | Macro F1 | Acc | macro F1 | Acc | macro F1 |
| Audio-only | 75.95 | 55.47 | 66.13 | 63.44 | 65.25 | 65.73 | - | - |
| Visual/Image-only | 73.47 | 50.23 | 64.95 | 62.74 | 64.33 | 65.60 | 56.50 | 45.70 |
| Text-only | 76.53 | 54.06 | 67.55 | 64.24 | 66.45 | 67.04 | 57.85 | 49.16 |
| Concatenation | 78.43 | 56.82 | 68.58 | 64.92 | 68.94 | 69.19 | 60.20 | 56.87 |
| Grad-Blending | 79.74 | 57.32 | 69.46 | 67.22 | 70.05 | 70.75 | 60.65 | 56.59 |
| OGM* | 81.05 | 57.59 | 70.34 | 67.80 | 71.43 | 71.90 | 61.30 | 58.56 |
| AGM* | 80.17 | 58.15 | 70.65 | 67.08 | 71.61 | 72.41 | 61.80 | 58.66 |
| PMR* | 79.45 | 57.53 | 68.35 | 65.82 | 69.68 | 70.41 | 60.35 | 57.25 |
| MMPareto* | 81.34 | 57.16 | 70.46 | 67.53 | 71.89 | 72.38 | 61.85 | 57.95 |
| D&R* | 80.61 | 56.63 | 71.07 | 67.93 | 70.88 | 71.64 | 61.35 | 54.60 |
| ARL* | 80.90 | 56.88 | 70.77 | 67.84 | 71.24 | 71.57 | 61.60 | 59.08 |
| GOAL | **81.78** | **58.32** | **71.23** | **68.22** | **72.90** | **73.61** | **62.30** | **60.51** |

A.9   PROOF OF THE ORTHOGONALITY OF EQU 9-10

Take Eq.9 as an example. We can prove it by showing that the dot product of $g^{(1)''}$ and $g^{(2)'}$ is zero.

**Proof:**

1. Compute the dot product: $g^{(1)''} \cdot g^{(2)'}$

2. Substitute the definition of $g^{(1)''}$:

$$= \left(g^{(1)'} - l_{g^{(1)} \to (2)}\right) \cdot g^{(2)'}$$

3. Expand the dot product:

$$= \left(g^{(1)'} \cdot g^{(2)'}\right) - \left(l_{g^{(1)} \to (2)} \cdot g^{(2)'}\right)$$

4. Substitute the definition of $l_{g^{(1)} \to (2)}$ (for clarity, ignore $\epsilon$):

$$= \left(g^{(1)'} \cdot g^{(2)'}\right) - \left(\left(\frac{g^{(1)'} \cdot g^{(2)'}}{\|g^{(2)'}\|^2} g^{(2)'}\right) \cdot g^{(2)'}\right)$$

5. The term inside the parentheses is a scalar, so we can regroup:

$$= \left(g^{(1)'} \cdot g^{(2)'}\right) - \left(\frac{g^{(1)'} \cdot g^{(2)'}}{\|g^{(2)'}\|^2}\right) \left(g^{(2)'} \cdot g^{(2)'}\right)$$

6. By definition, $\left(g^{(2)'} \cdot g^{(2)'}\right) = \|g^{(2)'}\|^2$.

7. The numerator and denominator cancel out:

$$= \left(g^{(1)'} \cdot g^{(2)'}\right) - \left(g^{(1)'} \cdot g^{(2)'}\right) = 0$$

By mathematical construction, the updated gradient $g^{(1)''}$ is necessarily orthogonal to $g^{(2)'}$. This operation removes the conflicting component by subtracting the projection of $g^{(1)'}$ onto $g^{(2)'}$, leaving only orthogonal (neutral) or synergistic (co-directional) components.

## A.10 THE EFFECT OF DIFFERENT VALUES OF T ON THE RESULTS IN EQU 7

Table 7: Accuracy with different T values.

| CREMA-D | | KS | | AVE | |
|---|---|---|---|---|---|
| T | Acc | T | Acc | T | Acc |
| 0.2 | 76.08 | 0.2 | 74.35 | 0.2 | 70.15 |
| 0.4 | 77.96 | 0.4 | 74.49 | 0.4 | 71.23 |
| 0.6 | 77.42 | 0.6 | 74.63 | 0.6 | 70.23 |
| 0.8 | 77.82 | 0.8 | 74.91 | 0.8 | 71.98 |
| 1.0 | 78.23 | 1.0 | 75.37 | 1.0 | 72.14 |

## A.11 COMPUTATION TIME AND MEMORY CONSUMPTION ANALYSIS

Table 8: Computation time and memory consumption analysis for per sample.

| Dataset | CREMA-D | | KS | | AVE | |
|---|---|---|---|---|---|
| Methods | Base | GOAL | Base | GOAL | Base | GOAL |
| Average training time per sample (ms) | 15.67 | 17.41 | 19.68 | 20.47 | 21.72 | 23.28 |
| Average training time per epoch (min) | 1.73 | 1.93 | 7.60 | 7.28 | 1.22 | 1.30 |
| GPU usage (MiB) | 1089 | 1089 | 1093 | 1093 | 1093 | 1093 |

## A.12 ANALYSIS OF RELEVANT INDICATORS FOR TSNE

The feature visualization in Figure 3 is not particularly clear visually, so we need to analyze the relevant clustering metrics.

Four classic clustering evaluation metrics are used to verify the optimization effect of the GOAL module: silhouette score (measures intra-cluster compactness and inter-cluster separation, closer to 1 is better), davies bouldin index (evaluates clustering discrimination, smaller value is better),

normalized mutual info (quantifies consistency between clustering and true labels, closer to 1 is better), and adjusted rand index (measures similarity between clustering and true categories, closer to 1 is better). Experimental results show that the GOAL module significantly outperforms the Base model on all metrics: the silhouette score nearly doubles, the davies bouldin index decreases by 25%, and the normalized mutual info and adjusted rand index increase 18% and 23%. This is attributed to the Adaptive Leveraging mechanism and Gradient Orthogonalization, which effectively enhance feature discriminability and eliminate modal conflicts.

Table 9: Analysis of relevant indicators.

| Metrics/Methods | Base | GOAL |
|---|---|---|
| Silhouette score($\uparrow$) | 0.074 | **0.131** |
| Davies bouldin index ($\downarrow$) | 2.566 | **1.910** |
| Normalized mutual info ($\uparrow$) | 0.457 | **0.939** |
| Adjusted rand index ($\uparrow$) | 0.403 | **0.498** |

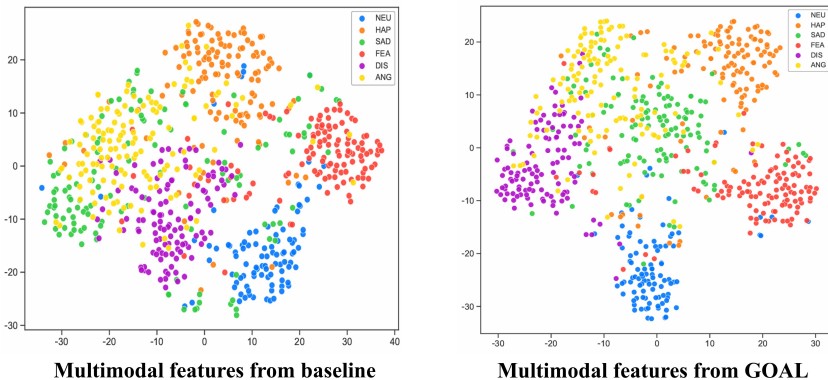

**Multimodal features from baseline**      **Multimodal features from GOAL**

Figure 6: Distribution of fused feature colored by different classes on the CREMA-D dataset by Tsne.

## A.13 Tracing the weights of gradients

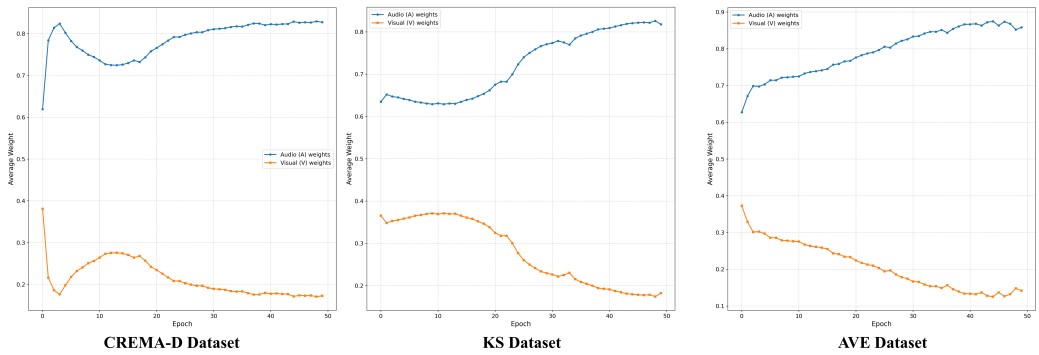

**CREMA-D Dataset**            **KS Dataset**            **AVE Dataset**

Figure 7: Tracing the weights of gradients

## A.14 The indirect effect of GOAL on gradients

According to our article, the GO and AL module only modifies the gradient $g$ generated by the multimodal loss, and does not modify the $G$ generated by the unimodal loss. However, at time $t$, the GOAL module will not affect $G$, but we can consider whether it has an effect at time $t+1$. In fact, the GOAL module actually has an indirect effect on $G$, as demonstrated below.

Let $\theta_t \in \mathbb{R}^d$ denote the model parameters at the $t-th$ iteration. In a multi-encoder architecture, the total loss function $\mathcal{L}_{\text{total}}$ consists of two parts:

$$\mathcal{L}_{\text{total}}(\theta) = \mathcal{L}_{mm}(\theta) + \mathcal{L}_s(\theta)$$

Where $\mathcal{L}_{mm}(\theta)$ is the loss of the multimodal task. $\mathcal{L}_s(\theta) = \sum_{m=1}^{M} \mathcal{L}^{(m)}(\theta)$ is the sum of the losses of all unimodal tasks.

Define the original gradient vectors at time $t$ : Multimodal gradient: $g_t = \nabla_\theta \mathcal{L}_{mm}(\theta_t)$, Unimodal gradient: $G_t = \nabla_\theta \mathcal{L}_s(\theta_t)$.

In standard Stochastic Gradient Descent (SGD), parameter updates follow the linear superposition of gradients. Let the learning rate be $\eta$, then the update rule of the Baseline method is:

$$\theta_{t+1}^{\text{base}} = \theta_t - \eta(g_t + G_t)$$

The GOAL framework introduces a nonlinear gradient correction operator $\Phi(\cdot)$, which acts only on the multimodal gradient $g_t$. GOAL generates the corrected gradient $\tilde{g}_t$ via the AL and GO modules:

$$\tilde{g}_t = \Phi_{GOAL}(g_t, \mathcal{D}_t) = \text{GO}(\text{AL}(g_t))$$

Thus, the parameter update rule under the GOAL framework becomes:

$$\theta_{t+1}^{goal} = \theta_t - \eta(\tilde{g}_t + G_t)$$

Let $\Delta v_t$ be the "correction vector" applied by the GOAL algorithm (relative to the Baseline algorithm) to the multimodal gradient at time $t$:

$$\Delta v_t = \tilde{g}_t - g_t$$

This leads to the difference in parameter updates:

$$\Delta\theta_{t+1} = \theta_{t+1}^{goal} - \theta_{t+1}^{base} = [\theta_t - \eta(\tilde{g}t + G_t)] - [\theta_t - \eta(g_t + G_t)]$$
$$\Delta\theta_{t+1} = -\eta(\tilde{g}_t - g_t) = -\eta\Delta v_t$$

This indicates that GOAL causes the parameters to move in the direction of $-\Delta v_t$ by a step size proportional to the learning rate.

We aim to examine the difference in the gradient of the unimodal loss function $\mathcal{L}_s$ at time $t + 1$. Hessian matrix at $\theta_t$ of $\mathcal{L}_s$ is:

$$\nabla\mathcal{L}_s(\theta_t + \delta) \approx \nabla\mathcal{L}_s(\theta_t) + \mathbf{H}_s(\theta_t)\delta + O(|\delta|^2)$$

We calculate the approximate single-modal gradients of the Baseline and GOAL at time $t + 1$ respectively:

1.Baseline Case: The update step size is: $\delta_{base} = -\eta(g_t + G_t)$, and $G_{t+1}^{base} \approx G_t + \mathbf{H}_s(\theta_t)[-\eta(g_t + G_t)]$.

2.GOAL Case: The update step size is: $\delta_{goal} = -\eta(\tilde{g}_t + G_t)$, and $G_{t+1}^{goal} \approx G_t + \mathbf{H}_s(\theta_t)[-\eta(\tilde{g}_t + G_t)]$.

Now, we calculate the difference between them, and the result is the indirect influence $\Delta G_{\text{indirect}}$ of the GOAL module on the single-modal gradient:

$$\Delta G_{indirect} = G_{t+1}^{goal} - G_{t+1}^{base}$$
$$\Delta G_{indirect} \approx G_t - \eta\mathbf{H}_s(\theta_t)(\tilde{g}_t + G_t) - G_t - \eta\mathbf{H}_s(\theta_t)(g_t + G_t)$$

$$\Delta G_{indirect} \approx -\eta \mathbf{H}_s(\theta_t)\tilde{g}_t + \eta \mathbf{H}s(\theta_t)g_t$$
$$\Delta Gindirect \approx -\eta \mathbf{H}_s(\theta_t)(\tilde{g}_t - g_t)$$

Substitute $\Delta v_t = \tilde{g}_t - g_t$, we obtain the final proof conclusion:

$$\Delta G_{indirect} \approx -\eta \mathbf{H}_s(\theta_t)\Delta v_t$$

From the above proof, we can see that the effect of GOAL on $G$ occurs at time $t+1$. Therefore, GOAL co-influences both $g$ and $G$.

### A.15 A PRELIMINARY STUDY ON THE INFLUENCE OF GOAL ON $g$ AND $G$

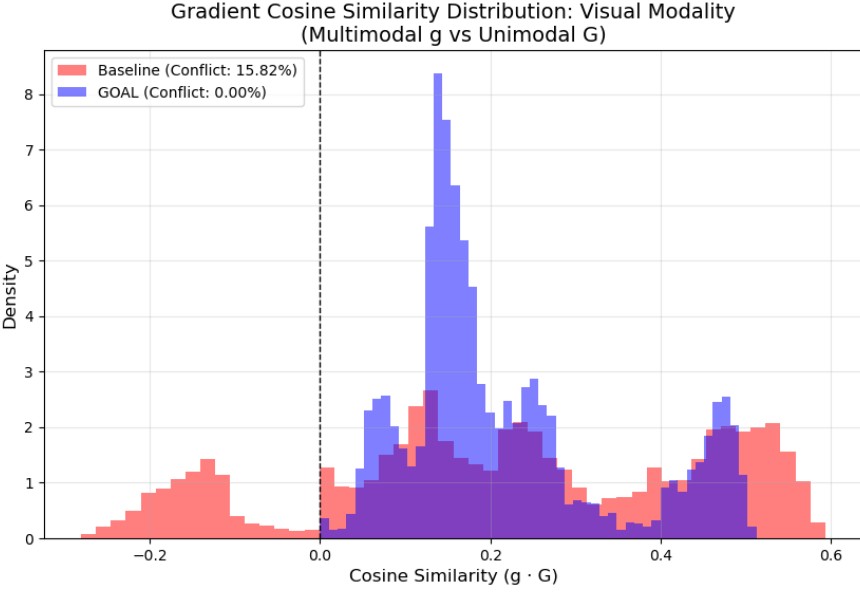

Figure 8: The final model used to compute the loss and the corresponding gradients $g$ and $G$ on the training set, and to check for any conflicts.

To verify whether GOAL effectively mitigates conflicts between $g$ and $G$, we computed the gradients and their cosine similarities across the entire training set using both the baseline model and the model trained with GOAL. As illustrated in Figure 8 (taking the visual modality as a representative example), the gradients $g$ and $G$ in the GOAL-trained model exhibit a high degree of alignment, demonstrating that our method naturally resolves optimization conflicts.

### A.16 COMPARISON OF CONVERGENCE SPEED BETWEEN GOAL AND BASELINE METHODS

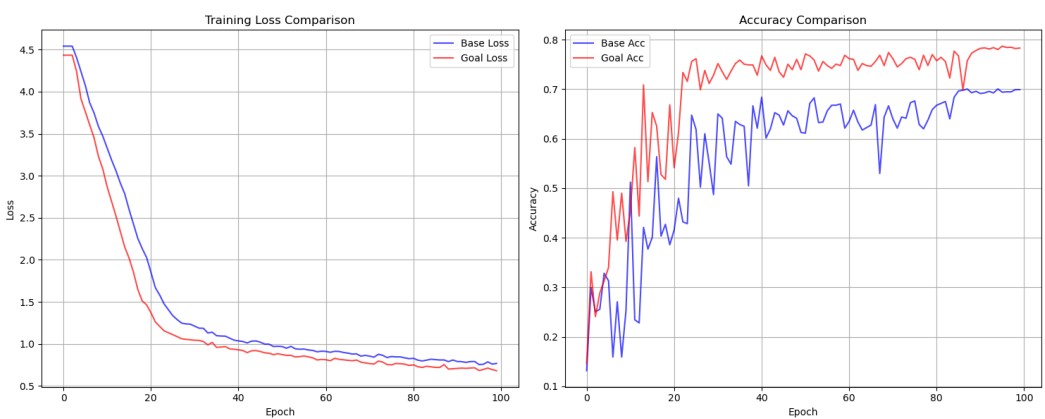

Figure 9: Comparison of convergence speed between GOAL and baseline methods

