# OpenReview forum: "GOAL: Balance Multimodal Learning with Gradient Orthogonalization and Adaptive Leveraging"
_ICLR.cc/2026/Conference — Submitted to ICLR 2026_

### Official Review · Reviewer_PpR3 · 2025-10-27

**Soundness:** 3
**Presentation:** 3
**Contribution:** 3
**Rating:** 6
**Confidence:** 3

**Summary:**

This paper addresses two key problems that hinder effective multimodal learning: (1) inconsistent gradient magnitudes across modalities, and (2) opposing gradient directions within a unimodal encoder. The authors argue that for better fusion, it's crucial to consider both within-modality representation strength and cross-modality compatibility.

Their main contribution is GOAL (Gradient Orthogonalization and Adaptive Leveraging), a parameter-free, plug-and-play gradient modification method with two components: (1) Adaptive Leveraging (AL): Dynamically re-weights gradient magnitudes for each modality based on prediction entropy, which serves as a variance estimator; (2) Gradient Orthogonalization (GO): Resolves conflicting gradients by projecting them to ensure synergistic updates and produce compatible multimodal features.

The paper demonstrates through extensive experiments that GOAL consistently and significantly outperforms existing state-of-the-art methods across various modalities and frameworks.

**Strengths:**

1. Proposes a novel multimodal optimization method, GOAL, which simultaneously addresses within-modality representation enhancement and cross-modality compatibility.

2. The proposed Adaptive Leveraging and Gradient Orthogonalization modules are well-designed with clear theoretical justification: AL adjusts gradient magnitudes based on prediction uncertainty, and GO alleviates modality conflicts through conflicting gradient projection.

3. GOAL is parameter-free, gradient-based, and can be used as a plug-and-play module.

4. Experiments are thorough and comprehensive, validating the effectiveness of the method.

**Weaknesses:**

1. The paper provides limited analysis on the quantification of gradient conflicts under different tasks, especially for complex multimodal scenarios.

2. The paper cites the analysis in [1], which suggests that “modality contribution is inversely proportional to prediction variance” to achieve optimal information fusion. Intuitively, if a modality has high prediction uncertainty (high variance), it is less reliable for the final decision and should be assigned a lower weight; conversely, a low-variance modality should receive a higher weight. However, does high uncertainty necessarily indicate low shared information across modalities? In other words, does high uncertainty also imply a low contribution?

[1] Wei S, Luo C, Luo Y. Improving Multimodal Learning via Imbalanced Learning, ICCV 2025.

**Questions:**

1. Does high uncertainty necessarily indicate low shared information across modalities? In other words, does high uncertainty also imply a low contribution?

2. Some works such as KuDA [2] and EMOE [3] also investigate modality contributions. What are the relative advantages and limitations of GOAL compared to these methods?

[2] Knowledge-Guided Dynamic Modality Attention Fusion Framework for  Multimodal Sentiment Analysis, EMNLP 2024.
[3] EMOE: Modality-Specific Enhanced Dynamic Emotion Experts, CVPR 2025

---

> ### Author Response · Authors · 2025-11-20
> **Part 1**
>
> **Weakness 1 :** The paper provides limited analysis on the quantification of gradient conflicts under different tasks, especially for complex multimodal scenarios.
>
> **Answer:** In Figure 2a in our main paper, we have provided the histogram of cosine similarity between gradients of the two modalities with the baseline method, where the gradient conflict can be observed, leading to suboptimal fusion performance (Figure 2b). For better understanding, we also exhibit the corresponding histogram after applying our proposed GOAL in Figure 5 in the Appendix A.6, which will be updated to Figure 2 in the final version. Clearly, there are much less conflicts after GOAL modelling. In Figure 2b, we provide another quantification manner, correlation coefficient, to illustrate the relationship between different modal features. The higher correlation coefficient values, the more compatible features, then the better fusion performances.
>
> **Weakness 2 \& Question 1 :** Does high uncertainty necessarily indicate low shared information across modalities? In other words, does high uncertainty also imply a low contribution?
>
> **Answer:** We thank the reviewer for this profound question. This is a critical point. Our theory does not establish a relationship between "uncertainty" and "quantity of information," but rather focuses on **"prediction reliability"**. A modality with **high uncertainty (high variance)** corresponds to an **unreliable prediction**; therefore, it should be assigned a lower contribution in the final fusion decision. This principle is not heuristic but stems from a rigorous mathematical derivation of the optimal fusion strategy.
>
> Allow us to explain mathematically (combining insights from [1]). The detailed proof is updated in Appendix A.7, here are the key points:
>
> For our model architecture, the output is
>
> $$f(x_{i})=W_{0}\cdot z_{0}+b_{0}+W_{1}\cdot z_{1}+b_{1}$$
>
> Here, $x$ is the input, $w$ and $b$ represent the parameters of the classification head, and z respectively stands for the features extracted by different single-modal encoders.
>
> The optimization dependency coefficient $ d $ is defined as follows,
>
> $$
> d = \frac{\text{d}^{m_0}}{\text{d}^{m_1}} = \frac{ \exp(s_i^{m_0}(y_i)) / \sum_{j=1}^M \exp(s_i^{m_0}(j)) }{ \exp(s_i^{m_1}(y_i)) / \sum_{j=1}^M \exp(s_i^{m_1}(j)) }
> $$
>
> Previous work has focused on making $d$ equal to 1, but in [1] it was found that increasing the gradient of the dominant modality by a factor of 5 actually improves performance. Therefore, the best optimization dependency coefficient is not 1. And it should satisfy the following formula,
>
> $$
> \begin{cases}
> \min_{w_0, w_1} \, g = \frac{1}{N} \sum_{i=1}^{N} L(f(x_i), y_i) \\\\
> f(x_i) = w_0 s_i^{m_0} + w_1 s_i^{m_1}
> \end{cases}
> $$
>
> where $w_0 > 0$ and $w_1 > 0$ denotes the contribution of modality $m_0$ and $m_1$, respectively, and $w_0 + w_1 = 1$.
>
> According to the bias-variance decomposition, the error between prediction and groundtruth can be rewritten as follows,
>
> $$
> \begin{cases}
> g = (\text{Bias}(f(x), y))^2 + \text{Var}(f(x)) + \text{Var}(\epsilon) \\\\
> \text{Bias}(f(x), y) = E\left[ f(x) - y \right] \\\\
> \text{Var}(f(x)) = E\left[ f(x)^2 \right] - (E\left[ f(x) \right])^2
> \end{cases}
> $$
>
> We need to minimize $\text{Var}(f(x))$. For the term of $\text{Var}(f(x))$, we can convert it as follows:
>
> $$
> \text{Var}(f(x)) = w_0^2 \text{Var}(s^{m_0}) + w_1^2 \text{Var}(s^{m_1})
> $$
>
> Then, with the constraint $w_0 + w_1 = 1$, we obtain the optimal solution for $w_0$ and $w_1$ as follows:
>
> $$
> \begin{cases}
> w_0 = \frac{\text{Var}(s^{m_1})}{\text{Var}(s^{m_1}) + \text{Var}(s^{m_0})} \\\\
> w_1 = \frac{\text{Var}(s^{m_0})}{\text{Var}(s^{m_1}) + \text{Var}(s^{m_0})}
> \end{cases}
> $$
>
> Consequently, the relationship between the weight ratio $w_0/w_1$ and the variances can be expressed as:
>
> $$
> \frac{w_0}{w_1} = \frac{\text{Var}(s^{m_1})}{\text{Var}(s^{m_0})} = \frac{1 / \text{Var}(s^{m_0})}{1 / \text{Var}(s^{m_1})}
> $$
>
> In other words, to minimize the overall $\text{Var}(f(x))$, the model should assign a larger weight (contribution) to the modality with lower variance.
>
> [1] Wei, Shicai, Chunbo Luo, and Yang Luo. "Improving multimodal learning via imbalanced learning." Proceedings of the IEEE/CVF International Conference on Computer Vision. 2025.

---

> > ### Author Response · Authors · 2025-11-20
> > **Part 2**
> >
> > **Question 2 :** Some works such as KuDA and EMOE also investigate modality contributions. What are the relative advantages and limitations of GOAL compared to these methods?
> >
> > **Answer:** In summary, EMOE and KuDA are **feature-level** and **architecture-level** solutions acting on the **forward** propagation process. GOAL is a **gradient/optimizer-level** solution acting on the **backward** propagation process. Crucially, GOAL introduces **no extra learnable parameters** and is **plug-and-play**.
> >
> > EMOE: Uses a parameterizable routing network taking features as input to output fusion weights.
> >
> > KuDA: Uses a two-stage framework and decoder to compute "emotion ratios" to adjust feature representations.
> >
> > GOAL: Intercepts gradients from the shared loss. Before updating the unimodal encoders, it modifies gradient magnitude (via AL) and direction (via GO).
> >
> > GOAL addresses the optimization process itself (modal laziness and gradient conflict), which differs from feature-level methods.
> >
> > **Limitations \& Scope:**
> >
> > Limitations: GOAL's effect is indirect. It modifies gradients to encourage encoders to learn compatible features, rather than explicitly gating features in the forward pass. We prioritized a fundamental optimization approach without major architectural changes.
> >
> > Scope: GOAL utilizes the "multi-encoder" architecture common in modal imbalance research, where gradients from the shared loss are clearly separable. In more complex architectures where gradients are coupled and hard to separate, this remains unexplored. This aligns with our discussion with Reviewer 2DRZ and is part of our future work.

---

### Official Review · Reviewer_YYZM · 2025-10-31

**Soundness:** 3
**Presentation:** 2
**Contribution:** 2
**Rating:** 2
**Confidence:** 5

**Summary:**

The paper proposes Gradient Orthogonalization Adaptive Leveraging (GOAL) considering unimodal representation and cross-modal compatibility via reweights gradient magnitude using entropy prediction as variation estimator. Experiments on multiple datasets are conducted to show effectiveness of the proposed model. However, the main method considering modality laziness [1] and modality orthogonalization strategy [2] are both explored and proposed in previous methods, which quite reduce the novelty of the proposed method.
[1] Weiyao Wang, Du Tran, and Matt Feiszli. What makes training multi-modal classification networkshard? In Proceedings ofthe lEEE/CVF conference on computer vision and pattern recognition.pp.12695-12705,2020.
[2] Yake Wei and Di Hu. Mmpareto: Boosting multimodal learning with innocent unimodal assistance arXiv preprint arXiv:2405.17730.2024

**Strengths:**

1.	Proposed strategies are theoretically solid.
2.	Experiments are partially effective.

**Weaknesses:**

1.	Modality laziness and modality adjustment are both proposed by previous methods. The method seems incremental since the difference has not been presented or described.
2.	There are two issues mentioned in Abstract. The paper should discuss the proposed solutions for both issues more clearly.
3.	More theory for both AL and GR modules should be provided with more math motivation.
4.	The orthogonality in Equ 9-10 should be described.
5.	The reported metrics are not sufficient for different datasets.
6.	Unimodal performance in Table 3 should be reported.
7.	Weights of gradients should be traced for adaptive learning.

**Questions:**

1.	Since primary modality mostly has greater confidence, will the weight of its gradients remain larger than other modalities, which increased the modality imbalance degree?
2.	Why does visualization of GOAL embeddings in Figure 3 not show significant effectiveness compared with baseline?

---

> ### Author Response · Authors · 2025-11-20
> **Part 1**
>
> We thank reviewer for acknowledging that our proposed strategies are theoretically solid. The reviewer point out several weaknesses and questions, for which we provide our discussions below.
>
> **Weakness 1 :** Modality laziness and modality adjustment are both proposed by previous methods. The method seems incremental since the difference has not been presented or described.
>
> **Answer**: We respectfully disagree. As discussed in Section 2.3, Section 3.2 in the main paper, compared with [1,2] and for multi-modal learning, our novelty lies in 1) identifying the new factor, i.e., **the gradient direction in cross-modality compatibility**, in modifying gradients from *modality-shared loss*, and 2) design an fast yet effective strategies accordingly.
>
> In detail, following Eq. 5 in the main paper, the gradient of parameters $\theta^{(m)}$ can separated as two parts, one ($g^{(m)}$) from the modality-shared loss ($L_{\text{mm}}$) and another one ($G^{(m)}$) from the modality-specific loss ($L^{(m)}$), i.e.,
>
> $$ \nabla_{\theta^{(m)}} L_{total} = \frac{\partial \mathcal{L}_{mm}}{\partial f^{(m)}} \frac{\partial f^{(m)}}{\partial \theta^{(m)}} + \frac{\partial \mathcal{L}^{(m)}}{\partial f^{(m)}} \frac{\partial f^{(m)}}{\partial \theta^{(m)}} = (g^{(m)} + G^{(m)}) \frac{\partial f^{(m)}}{\partial \theta^{(m)}}$$
>
> **Comparison with Related Work:** While the problems of modal imbalance was first proposed in [1], the proposed solution as well as the following methods [3,4,5,6,7] focus exclusively on aligning the magnitudes of the gradients derived from $L_{\text{mm}}$ for each modality, i.e., $|g^{(1)}|$ v.s. $|g^{(2)}|$ (assuming $m\in \{1,2\}$ for simplicity), neglecting the directional alignment. Nevertheless, the directional alignment consider in [2] is between $g^{(m)}$ and $G^{(m)}$. Instead, our approach explicitly considers the directional alignment between the gradients $g^{(1)}$ and $g^{(2)}$.
>
> While our method do not explicitly regulate the direction of $g^{(m)}$ and $G^{(m)}$, through the mathematical analysis in A.14, our method may still facilitate the directional alignment between $g^{(m)}$ and $G^{(m)}$. As demonstrated in Fig. 8 in the appendix, the the conflict of gradients for visual modality in baseline is effectively resolved in our method.
>
> **(Novelty 1: Compared with existing methods of modifying the gradient direction: We identify a new factor for imbalance problem, which motivates us to propose a new method)**
> Beyond the alignment logic mentioned above, our core insight is identifying cross-modality compatibility as a critical yet overlooked factor for effective fusion. As validated by the CCA analysis in Fig. 2b, aligning the gradient directions across modalities ($g^{(1)}$ vs. $g^{(2)}$) significantly enhances feature compatibility. This directional alignment directly leads to the superior performance of GOAL compared to intra-modality alignment methods like Mmpareto (Table 1 \& 3).
>
>
> **(Novelty 2: Compared with existing methods of modifying the gradient magnitude: GOAL is simple but effective to calculate the balanced weight)**
> While previous works often rely on complex optimization for magnitude balancing, GOAL adopts a streamlined, contribution-based strategy. We dynamically re-weight gradient magnitudes using task-specific metrics (e.g., entropy for classification, variance for regression). This realization is simple yet effective, without introducing heavy computational overhead.
>
> [1] Weiyao Wang, Du Tran, and Matt Feiszli. What makes training multi-modal classification networkshard? In Proceedings ofthe lEEE/CVF conference on computer vision and pattern recognition.pp.12695-12705,2020.
>
> [2] Wei, Yake, and Di Hu. "Mmpareto: Boosting multimodal learning with innocent unimodal assistance." arXiv preprint arXiv:2405.17730 (2024).
>
> [3] Fan, Yunfeng, et al. "Pmr: Prototypical modal rebalance for multimodal learning." Proceedings of the IEEE/CVF Conference on Computer Vision and Pattern Recognition. 2023.
>
> [4] Li, Hong, et al. "Boosting multi-modal model performance with adaptive gradient modulation." Proceedings of the IEEE/CVF International Conference on Computer Vision. 2023.
>
> [5] Peng, Xiaokang, et al. "Balanced multimodal learning via on-the-fly gradient modulation." Proceedings of the IEEE/CVF conference on computer vision and pattern recognition. 2022.
>
> [6] Wei, Yake, et al. "Diagnosing and re-learning for balanced multimodal learning." European Conference on Computer Vision. Cham: Springer Nature Switzerland, 2024.
>
> [7] Wei, Shicai, Chunbo Luo, and Yang Luo. "Improving multimodal learning via imbalanced learning." Proceedings of the IEEE/CVF International Conference on Computer Vision. 2025.

---

> > ### Author Response · Authors · 2025-11-20
> > **Part 2**
> >
> > **Weakness 2 :** There are two issues mentioned in Abstract. The paper should discuss the proposed solutions for both issues more clearly.
> >
> > **Answer:** We have discussed the proposed solutions for both issues in Section 3.3 (AL component) and Section 3.4 (GO component) in the main paper, which correspondents to the two novelties mentioned above. Specifically,
> > For the first issue about opposing gradient directions in a uni-modal encoder, the GO component is developed by gradient orthogonalization.
> > For the second issue arises from inconsistent gradient magnitudes across modalities, the AL component is introduced by a variance-based leveraging mechanism. Grounded in the theory that optimal fusion weights are inversely proportional to variance, AL dynamically scales gradients by using prediction entropy as a proxy for model uncertainty.
> >
> > **Weakness 3 :** More theory for both AL and GR modules should be provided with more math motivation.
> >
> > **Answer:** We thank the reviewer for the suggestion and your acknowledgement that our  “proposed strategies are theoretically solid” (see your Strengths 1). Meanwhile, Reviewer PpR3 states that "well-designed with clear theoretical justification", and Revewier 2DRZ comments "with the theoretical analysis and experimental sections supporting each other".
> >
> > For your reference, we provide additional theory and mathematic motivation as follows (details added in Appendix A.7). But to address your concern most effectively, we still appreciate your clarification on what type of theory is needed.
> >
> > **About AL module**
> >
> > The research motivation stems from the findings [7] where the current "balanced" approaches for gradient magnitude control is sub-optimal; instead, the unbalanced ones which are proportional to variance within each modality are preferred. Specifically, assuming that the current input consists of two modalities, for the architecture of a multimodal model, our final model output is:
> >
> > $$f(x_{i})=W_{0}\cdot z_{0}+b_{0}+W_{1}\cdot z_{1}+b_{1}$$
> >
> > Here, $x$ is the input, $w$ and $b$ represent the parameters of the classification head, and z respectively stands for the features extracted by different single-modal encoders.
> >
> > The optimization dependency coefficient $ d $ is defined as follows,
> >
> > $$
> > d = \frac{\text{d}^{m_0}}{\text{d}^{m_1}} = \frac{ \exp(s_i^{m_0}(y_i)) / \sum_{j=1}^M \exp(s_i^{m_0}(j)) }{ \exp(s_i^{m_1}(y_i)) / \sum_{j=1}^M \exp(s_i^{m_1}(j)) }
> > $$
> >
> > Previous work has focused on making $d$ equal to 1, but in [7] it was found that increasing the gradient of the dominant modality by a factor of 5 actually improves performance. Therefore, the best optimization dependency coefficient is not 1. And it should satisfy the following formula,
> >
> > $$
> > \begin{cases}
> > \min_{w_0, w_1} \, g = \frac{1}{N} \sum_{i=1}^{N} L(f(x_i), y_i) \\\\
> > f(x_i) = w_0 s_i^{m_0} + w_1 s_i^{m_1}
> > \end{cases}
> > $$
> >
> > where $w_0 > 0$ and $w_1 > 0$ denotes the contribution of modality $m_0$ and $m_1$, respectively, and $w_0 + w_1 = 1$.
> >
> > According to the bias-variance decomposition, the error between prediction and groundtruth can be rewritten as follows,
> >
> > $$
> > \begin{cases}
> > g = (\text{Bias}(f(x), y))^2 + \text{Var}(f(x)) + \text{Var}(\epsilon) \\\\
> > \text{Bias}(f(x), y) = E\left[ f(x) - y \right] \\\\
> > \text{Var}(f(x)) = E\left[ f(x)^2 \right] - (E\left[ f(x) \right])^2
> > \end{cases}
> > $$
> >
> > We need to minimize $\text{Var}(f(x))$. For the term of $\text{Var}(f(x))$, we can convert it as follows:
> >
> > $$
> > \text{Var}(f(x)) = w_0^2 \text{Var}(s^{m_0}) + w_1^2 \text{Var}(s^{m_1})
> > $$
> >
> > Then, with the constraint $w_0 + w_1 = 1$, we obtain the optimal solution for $w_0$ and $w_1$ as follows:
> >
> > $$
> > \begin{cases}
> > w_0 = \frac{\text{Var}(s^{m_1})}{\text{Var}(s^{m_1}) + \text{Var}(s^{m_0})} \\\\
> > w_1 = \frac{\text{Var}(s^{m_0})}{\text{Var}(s^{m_1}) + \text{Var}(s^{m_0})}
> > \end{cases}
> > $$
> >
> > Consequently, the relationship between the weight ratio $w_0/w_1$ and the variances can be expressed as:
> >
> > $$
> > \frac{w_0}{w_1} = \frac{\text{Var}(s^{m_1})}{\text{Var}(s^{m_0})} = \frac{1 / \text{Var}(s^{m_0})}{1 / \text{Var}(s^{m_1})}
> > $$
> >
> > In other words, to minimize the overall $\text{Var}(f(x))$, the model should assign a larger weight (contribution) to the modality with lower variance.
> >
> > [7] Wei, Shicai, Chunbo Luo, and Yang Luo. "Improving multimodal learning via imbalanced learning." Proceedings of the IEEE/CVF International Conference on Computer Vision. 2025.

---

> > > ### Author Response · Authors · 2025-11-20
> > > **Part 3**
> > >
> > > **About GO module**
> > >
> > > Mathematical Motivation of the Gradient Orthogonalization (GO) Module:
> > > Feature Space Compatibility
> > >
> > > In the "multi-encoder" framework of GOAL (as shown in Figure 1 of the paper), each modality is equipped with an independent encoder that generates features. These features are then combined by a fusion module, and the shared modal fusion loss is computed. We must clarify the gradient object of interest in GOAL: $g_m$ denotes the gradient of the shared modal fusion loss $l_{\text{mm}}$ with respect to the feature output $f_m$ of the m-th modality. This $g_m$ is a vector in the feature space, and its mathematical meaning is: to minimize the shared modal fusion loss $l_{\text{mm}}$, the direction in which the fusion module expects the feature $f_m$ to "move" on its manifold. Therefore, "feature space compatibility" can be formally defined as: whether the corresponding feature gradients $g_m$ of all modalities are geometrically synergistic. This gives rise to our GO module.
> > >
> > > **Weakness 4 :** More description of orthogonality in Equ 9-10.
> > >
> > > **Answer:** We make a mathematical proof of the orthogonality in Eq. 9-10 as follows, and also add them in the Appendix A.9.
> > >
> > > Take Eq. 9 as an example. We can prove it by showing that the dot product of $g^{(1)''}$ and $g^{(2)'}$ is zero.
> > >
> > > **Proof:**
> > >
> > > 1. Compute the dot product: $g^{(1)''} \cdot g^{(2)'}$
> > >
> > > 2. Substitute the definition of $g^{(1)''}$:
> > > $$
> > > = \left( g^{(1)'} - l_{g^{(1)\rightarrow(2)}} \right) \cdot g^{(2)'}
> > > $$
> > >
> > > 3. Expand the dot product:
> > > $$
> > > = \left( g^{(1)'} \cdot g^{(2)'} \right) - \left( l_{g^{(1)\rightarrow(2)}} \cdot g^{(2)'} \right)
> > > $$
> > >
> > > 4. Substitute the definition of $l_{g^{(1)\rightarrow(2)}}$ (for clarity, ignore $\epsilon$):
> > > $$
> > > = \left( g^{(1)'} \cdot g^{(2)'} \right) - \left( \left( \frac{g^{(1)'} \cdot g^{(2)'}}{\|g^{(2)'}\|^2} g^{(2)'} \right) \cdot g^{(2)'} \right)
> > > $$
> > >
> > > 5. The term inside the parentheses is a scalar, so we can regroup:
> > > $$
> > > = \left( g^{(1)'} \cdot g^{(2)'} \right) - \left( \frac{g^{(1)'} \cdot g^{(2)'}}{\|g^{(2)'}\|^2} \right) \left( g^{(2)'} \cdot g^{(2)'} \right)
> > > $$
> > >
> > > 6. By definition, $\left( g^{(2)'} \cdot g^{(2)'} \right) = \|g^{(2)'}\|^2$.
> > >
> > > 7. The numerator and denominator cancel out:
> > > $$
> > > = \left( g^{(1)'} \cdot g^{(2)'} \right) - \left( g^{(1)'} \cdot g^{(2)'} \right) = 0
> > > $$
> > >
> > > **Weakness 5 :** The reported metrics are not sufficient for different datasets.
> > >
> > > **Answer:** Our paper is built upon and extends prior work [1-7] and cover all metrics used across the related tasks.  In detail, we maintained consistency with previous work, using the Acc and F1-Macro as the metrics. Additionally, for the validation datasets, we consider three different diversities and used three trimodal datasets (vision, language, and audio) and one image-text modality dataset, significantly more than previous work. To ensure our response is as targeted as possible, we highly recommend the reviewer to give more clear and specific instructions for further discussion, and we will gladly provide that analysis promptly.
> > >
> > > [1] Weiyao Wang, Du Tran, and Matt Feiszli. What makes training multi-modal classification networkshard? In Proceedings ofthe lEEE/CVF conference on computer vision and pattern recognition.pp.12695-12705,2020.
> > >
> > > [2] Wei, Yake, and Di Hu. "Mmpareto: Boosting multimodal learning with innocent unimodal assistance." arXiv preprint arXiv:2405.17730 (2024).
> > >
> > > [3] Fan, Yunfeng, et al. "Pmr: Prototypical modal rebalance for multimodal learning." Proceedings of the IEEE/CVF Conference on Computer Vision and Pattern Recognition. 2023.
> > >
> > > [4] Li, Hong, et al. "Boosting multi-modal model performance with adaptive gradient modulation." Proceedings of the IEEE/CVF International Conference on Computer Vision. 2023.
> > >
> > > [5] Peng, Xiaokang, et al. "Balanced multimodal learning via on-the-fly gradient modulation." Proceedings of the IEEE/CVF conference on computer vision and pattern recognition. 2022.
> > >
> > > [6] Wei, Yake, et al. "Diagnosing and re-learning for balanced multimodal learning." European Conference on Computer Vision. Cham: Springer Nature Switzerland, 2024.
> > >
> > > [7] Wei, Shicai, Chunbo Luo, and Yang Luo. "Improving multimodal learning via imbalanced learning." Proceedings of the IEEE/CVF International Conference on Computer Vision. 2025.

---

> > > > ### Author Response · Authors · 2025-11-20
> > > > **Part 4**
> > > >
> > > > **Weakness 6 :** Unimodal performance in Table 3 should be reported.
> > > >
> > > > **Answer:** We have added unimodal performance in Table 3 (see revised paper). Here we also show it.
> > > >
> > > > **Table: Comparison of different methods on Audio-Visual-Text and Image-Text datasets.** The "*" means we extend the original method to a trimodal version. Best results are **bold**.
> > > >
> > > >  | Method | MOSI (Acc) | MOSI (F1) | MELD (Acc) | MELD (F1) | IEMOCAP (Acc) | IEMOCAP (F1) | Hateful (Acc) | Hateful (F1) |
> > > >  | :--- | :---: | :---: | :---: | :---: | :---: | :---: | :---: | :---: |
> > > >  | Audio-only | 75.95 | 55.47 | 66.13 | 63.44 | 65.25 | 65.73 | - | - |
> > > >  | Visual/Image-only | 73.47 | 50.23 | 64.95 | 62.74 | 64.33 | 65.60 | 56.50 | 45.70 |
> > > >  | Text-only | 76.53 | 54.06 | 67.55 | 64.24 | 66.45 | 67.04 | 57.85 | 49.16 |
> > > >  | Concatenation | 78.43 | 56.82 | 68.58 | 64.92 | 68.94 | 69.19 | 60.20 | 56.87 |
> > > >  | Grad-Blending | 79.74 | 57.32 | 69.46 | 67.22 | 70.05 | 70.75 | 60.65 | 56.59 |
> > > >  | OGM* | 81.05 | 57.59 | 70.34 | 67.80 | 71.43 | 71.90 | 61.30 | 58.56 |
> > > >  | AGM* | 80.17 | 58.15 | 70.65 | 67.08 | 71.61 | 72.41 | 61.80 | 58.66 |
> > > >  | PMR* | 79.45 | 57.53 | 68.35 | 65.82 | 69.68 | 70.41 | 60.35 | 57.25 |
> > > >  | MMPareto* | 81.34 | 57.16 | 70.46 | 67.53 | 71.89 | 72.38 | 61.85 | 57.95 |
> > > >  | D\&R* | 80.61 | 56.63 | 71.07 | 67.93 | 70.88 | 71.64 | 61.35 | 54.60 |
> > > >  | ARL* | 80.90 | 56.88 | 70.77 | 67.84 | 71.24 | 71.57 | 61.60 | 59.08|
> > > >  | **GOAL** | **81.78** | **58.32** | **71.23** | **68.22** | **72.90** | **73.61** | **62.30** | **60.51** |
> > > >
> > > > **Weakness 7 :** Weights of gradients should be traced for adaptive learning.
> > > >
> > > > **Answer:** We have added a figure about tracing the weights of gradients in Appendix A.13.
> > > >
> > > > **Question 1 :** Since primary modality mostly has greater confidence, will the weight of its gradients remain larger than other modalities, which increased the modality imbalance degree?
> > > >
> > > > **Answer:** Your thought about “primary modality mostly has greater confidence, which increased the modality imbalance degree” is correct. This theory is primary developed in [7] oriented from asymmetric learning (not imbalanced learning), and we provide mathematical proof in “response of weakness 3” and Appendix A.7. Intuitively, the imbalanced problem in multimodal learning arises from the feature fusion with same weights. Asymmetric learning theory indicates that when fusing features, we need to consider the contribution of different modalities and reduce the weights of modality with low contribution. Similar idea is widely applied in multimodal learning, such as references provided by Reviewer PpR3.
> > > >
> > > > Besides, we should point out that although AL component in GOAL uses the theory developed in [7], but AL uses a more simple and effective realization.
> > > >
> > > > **Question 2 :** Why does visualization of GOAL embeddings in Figure 3 not show significant effectiveness compared with baseline?
> > > >
> > > > **Answer:** Figure 3 can illustrate the effectiveness of GOAL, but the improvement is not significant. This is because that during dimensionality reduction, t-SNE does not consider the **global structure** of the original features [8]. To solve this problem for better visualization, **UMAP** [8] is proposed. According to your suggestion, we replace current Figure 3 with the **UMAP visualization** (see revised paper; we remove tsne figure to Appendix), which shows significant effectiveness compared with baseline.
> > > >
> > > > No matter for UMAP or t-SNE, they use dimensionality reduction, which may destroy the structure of original features. Therefore, in order to quantitatively evaluate the discriminant of features, we use different clustering metrics as follows. Clearly, GOAL achieves better clustering metrics than baseline, which indicates that the features learned by GOAL is more discriminant. We also add this table in Appendix A.12.
> > > >
> > > > **Table: Analysis of relevant indicators.**
> > > >
> > > >  | Metrics/Methods | Base | GOAL |
> > > >  | :--- | :---: | :---: |
> > > >  | Silhouette score ($\uparrow$) | 0.074 | **0.131** |
> > > >  | Davies bouldin index ($\downarrow$) | 2.566 | **1.910** |
> > > >  | Normalized mutual info ($\uparrow$) | 0.457 | **0.939** |
> > > >  | Adjusted rand index ($\uparrow$) | 0.403 | **0.498** |
> > > >
> > > > [7] Wei, Shicai, Chunbo Luo, and Yang Luo. "Improving multimodal learning via imbalanced learning." Proceedings of the IEEE/CVF International Conference on Computer Vision. 2025.
> > > >
> > > > [8] McInnes, Leland, John Healy, and James Melville. "Umap: Uniform manifold approximation and projection for dimension reduction." arXiv preprint arXiv:1802.03426 (2018).

---

> ### Author Response · Authors · 2025-11-28
>
> Dear Reviewer YYZM,
>
> Thank you again for your constructive feedback and for acknowledging the theoretical solidity of our work.
>
> We have submitted a detailed response addressing your questions regarding theoretical motivations, novelty distinctions, and additional experimental metrics. Specifically, we have included new mathematical proofs and updated visualizations (UMAP and clustering metrics) in the revised paper to support our claims.
>
> We would be very grateful if you could take a moment to review our response. We are happy to provide further clarifications if any concerns remain.
>
> Best regards, Authors

---

### Official Review · Reviewer_YvAS · 2025-11-01

**Soundness:** 2
**Presentation:** 2
**Contribution:** 2
**Rating:** 4
**Confidence:** 4

**Summary:**

The paper proposes GOAL (Gradient Orthogonalization and Adaptive Leveraging), a parameter-free gradient modification framework to address imbalance issue in multimodal learning. Specifically, the method consists of two components: Adaptive Leveraging (AL), which adjusts the magnitude of gradients based on uncertainty estimation, and Gradient Orthogonalization (GO), which resolves conflicts between gradients from different modalities.

The paper demonstrates the efficacy of GOAL through experiments on multiple multimodal datasets. The proposed method is a plug-and-play module that can be integrated into existing multimodal learning frameworks, making it easy to be applied.

**Strengths:**

- The method is evaluated on multiple multimodal datasets, demonstrating consistent improvements in both accuracy and macro F1 score compared to related methods.
- GOAL is a plug-and-play module, making it easy to integrate into existing multimodal frameworks.
- The writing is easy to follow.

**Weaknesses:**

- The paper suggests that the gradient direction alignment between unimodal encoders is crucial. However, it remains unclear why aligning gradients in separate unimodal encoders (whose parameters are not shared) would significantly benefit model performance. Unlike shared parameters, which face potential gradient conflicts, the gradients of separate encoders are expected to be independent. The authors should clarify the theoretical and practical motivation behind aligning these gradients.
- Suppose it is meaningful to align the direction of different unimodal encoders. The paper presents gradient conflicts in Figure 2(a) between two modalities, but it does not clearly demonstrate whether the GOAL method effectively mitigates this conflict. After applying GOAL, is there a noticeable change in the gradient conflict?
- The use of the AL module to adaptively adjust gradient magnitudes across modalities is based on the assumption that the magnitudes of gradients from different modalities may vary due to their differing contribution to the learning objective. Further theoretical and empirical evidence is needed to justify this approach. Why is it crucial to address this difference in magnitude for improving fusion performance?

**Questions:**

Please check the above section.

---

> ### Author Response · Authors · 2025-11-20
> **Part 1**
>
> We thank reviewer for acknowledging the consistent improvements of GOAL on multiple multimodal datasets with its easy-to-integrate property. The reviewer asks about the motivation of aligning the gradient’s direction between unimodal encoders, the proof of change in the gradient conflict, and the importance of addressing the difference in magnitude for improving fusion performance. We provide our thoughts below.
>
>
> **Weakness 1 :** Why aligning gradients in separate unimodal encoders would significantly benefit model performance.
>
> **Clarification:** I think you might be mistakenly believing that we're dealing with the gradients corresponding to the model parameters. In fact, we're dealing with the gradients corresponding to the features used in downstream tasks. Modifying this gradient will affect every part of the model. Specifically, as defined in Eq. 5 in the main paper, the "gradient with respect to unimodal encoders" (Line 49) is $g^{(m)}$ and specifically refers to the gradient of the loss with respect to the feature outputs by uni-modal encoders. The term is named to be in line with established terminology in related research [1,2] and considers the fact that this gradient influences the update of all parameters within the unimodal encoder through back-propagation. Then, the following explanation regarding the "direction'' and "magnitude'' that we analyze is precisely the direction of these gradient vectors $g^{(m)}$, instead of the gradient corresponding to specific parameters.
>
> **Answer:** For one of the common frameworks in multimodal learning (Figure 1 in the paper) having multiple encoders with both **modality-specific** ($L^{(1)},…, L^{(M)} $) and **modality-shared** losses ($L_{\text{mm}}$), we point that GOAL does **not** align **gradients from both of them ($L^{(m)}+L_{\text{mm}}$)** in separate unimodal encoders, but just aligns the **gradient from the modality-shared loss $L_{\text{mm}}$**. We provide our motivation behind from the following two aspects.
>
> (*Intuitive motivation and mathematical derivation in Eq. 5: improve the cross-modality compatibility for efficient fusion*)
> As the description in 3.1, based on the above framework, the gradient of each encoder can be denoted as (also shown in Eq. 5 in the paper):
>
> $$ \nabla_{\theta^{(m)}} L_{total} = \frac{\partial \mathcal{L}_{mm}}{\partial f^{(m)}} \frac{\partial f^{(m)}}{\partial \theta^{(m)}} + \frac{\partial \mathcal{L}^{(m)}}{\partial f^{(m)}} \frac{\partial f^{(m)}}{\partial \theta^{(m)}} = (g^{(m)} + G^{(m)}) \frac{\partial f^{(m)}}{\partial \theta^{(m)}}$$
>
> Rather than aligning the gradient $G^{(m)} +g^{(m)} $ ,GOAL just aligns the gradient $g^{(m)} $ from modality-shared loss. Firstly, for the gradient from modality-specific loss $ G^{(m)} $, as you claimed, this gradients of separate encoders are expected to be independent. In our paper, we think this is to ensure the within-modality representation. Secondly, $g^{(m)} $ is derived from the modality-shared loss, we hope to aligning them among different modalities for better cross-modality compatibility.
>
> (*Experimental validation in Figure 2: higher correlation brings better performance after aligning gradient from modality-shared loss*)
> In Figure 2b, we apply the Canonical Correlation Analysis (CCA) to track the evolution of the feature correlation coefficient between different modalities during training. We find that i) baseline has the lowest correlation; ii) compared methods (OGM, ARL, and MmPareto) have the medium correlation; iii) GOAL have the largest correlation, which illustrates that GOAL achieves more compatible features. To go further, we add GO module (aligning gradient) to baseline, OGM, and ARL, the performance has increased, which denotes that the higher correlation does indeed lead to better performance.
>
> The above discussion is one of the important contributions or novelties of GOAL to alleviate the modality imbalance problem compared with all existing methods. Some of the above discussion is also included in the Section 3.2 to improve the quality of the articles.

---

> > ### Author Response · Authors · 2025-11-20
> > **Part 2**
> >
> > **Weakness 2 :** Figure 2a) just show the strong gradient conflict of baseline method. Is there a noticeable change in the gradient conflict after applying GOAL?
> >
> > **Answer:** We add a figure to visualize the situation of gradient conflict after applying GOAL in Appendix A.6. Clearly, it demonstrates that the GOAL method effectively mitigates this conflict.
> >
> > **Weakness 3 :** Why is it crucial to address this difference in magnitude for improving fusion performance?
> >
> > **Answer:** Recapping the Eq. 5 in the main paper, after having obtained the gradient of the parameters $\theta^{(m)} $, the SGD-like algorithm can be applied as:
> >
> > $$ \boldsymbol{\theta}^{(m)} \leftarrow \boldsymbol{\theta}^{(m)} - r \left( \|G^{(m)}\| \cdot \mathbf{b} + \|g^{(m)}\| \cdot \mathbf{a} \right) $$
> >
> > where, $r$ is the learning rate; $a$ and $b$ are the unit vectors whose direction are the same with $g^{(m)} $ and $G^{(m)} $, respectively. Obviously, with the same learning rate for all encoders, different magnitudes of the gradient, $g^{(m)} $ and $G^{(m)} $ , will affect the update speed of different encoders. If we do not address such difference, due to the high heterogeneity of data from different modalities, the following situations are very likely to occur: one encoder is overfitting, but another one is still underfitting. In our proposed GOAL, like what we do for aligning gradient direction, the proposed AL module is just adaptively adjust gradient magnitude for $g^{(m)} $. This strategy is similar with the previous works [1], but the adjusted weight in GOAL is more easy to estimate, which gives GOAL a plug-and-play property for more applications.
> >
> > **Summarization:** The answers for weakness 1 and 3 illustrate our contribution and novelty of GOAL compared with the existing works. We find that for alleviating the modality imbalance in “multi-encoder” architecture, an effective way is to align the direction and adjust gradient magnitude of $g^{(m)}$  to enhance the cross-modality compatibility, but do not modify   to maintain within-modality representations.
> >
> > [1] Wei, Shicai, Chunbo Luo, and Yang Luo. "Improving multimodal learning via imbalanced learning." Proceedings of the IEEE/CVF International Conference on Computer Vision. 2025.

---

> > ### Comment · Reviewer_YvAS · 2025-11-26
> > **Further concerns about gradient alignment**
> >
> > Thank you for the further explanation. First, I would like to confirm that the final gradient for each unimodal encoder comes from two sources: the unimodal loss and the multimodal loss. Accordingly, there are two gradients, unimodal gradient and multimodal gradient. And the gradients that the method attempts to align are the multimodal gradients across the unimodal encoders.
> >
> > If so, it indeed relates to my earlier concern, which I will further clarify here.
> >
> > The unimodal encoders are inherently heterogeneous because they process fundamentally different types of data. Even though they share the same learning objective (multimodal loss), the update directions they need may still be different. This is because each modality has its own input space and may require a different way to extract the same concept. From an intuitive perspective, differences among the multimodal gradients are natural.
> >
> > Given this, the proposed method that forces alignment of multimodal gradients across unimodal encoders lacks a strong justification for why these gradients should be aligned, even though it may bring some performance improvement.

---

> ### Author Response · Authors · 2025-11-27
> **Response to "Further concerns about gradient alignment"**
>
> Thank you for your further comment. We provide more explanations about your concern.
>
> **Clarification of the gradient alignment: the direction is NOT modified to be completely consistent but just remove the conflict part.**
>
> We totally agree with your opinion about "the unimodal encoders are inherently heterogeneous and differences among the multimodal gradients are natural".
> However, in our work, "*gradient alignment*" does NOT mean forcing gradient directions to be the same, but instead removing conflicting components [1].
> As shown in Figure 1 in the main paper, we eliminate the negative components in the gradients by changing the direction of $g^{(1)}$ and $g^{(2)}$ with obtuse angle ($\cos(g^{(1)}, g^{(2)})<0$) to be orthogonal ($\cos(g^{(1)}, g^{(2)})=0$), while keeping others ($\cos(g^{(1)}, g^{(2)}) \geq 0$) unchanged.
> Also, as shown in Figure 5 in the Appendix, after learning by GOAL, the cosine similarity between $g^{(1)}$ and $g^{(2)}$ are diverse from 0 to 1.
> Therefore, GOAL still **keeps the difference among the unimodal encoders**, but **just removes the conflict part** to ensure the **feature compatible** among different modalities.
>
> In multi-modal learning, when features from different modalities share similar semantics, their dot product in the shared embedding space is usually positive, even if features are concatenated and processed through a single linear layer, the weight vectors corresponding to each modality still have positive dot-product by default at the existing PyTorch initialization. Consequently, removing the conflicting gradients from multi-modal loss can enhance training acceleration and stability.
> This can be demonstrated in Figure 9 of Appendix A.16.
> Further, as demonstrated in A.15 in the appendix, even without explicit supervision, our method (GOAL) also eliminates the conflicting gradient components [1] from unimodal and multi-modal, which aligns with previous effort [2] and enhances training efficiency and robustness of the learned models.
>
> **Modification of the gradient: NOT for encoder's gradients from all layers but just for the gradients of the feature from the final layer.**
>
> Recapping the following equation:
>
> $$
> \nabla_{\theta^{(m)}} L_{total} = \frac{\partial \mathcal{L}_{mm}}{\partial f^{(m)}} \frac{\partial f^{(m)}}{\partial \theta^{(m)}} + \frac{\partial \mathcal{L}^{(m)}}{\partial f^{(m)}} \frac{\partial f^{(m)}}{\partial \theta^{(m)}} = (g^{(m)} + G^{(m)}) \frac{\partial f^{(m)}}{\partial \theta^{(m)}}.
> $$
>
> Instead of modifying the gradients of *parameters from all layers* in unimodal encode directly, GOAL just focuses on dealing with the gradients of *features from the final layer* ($g^{(m)}= \frac{\partial \mathcal{L}_{mm}}{\partial f^{(m)}} $).
>
> **Summary**
>
> Due to the above two well-designed process in GOAL, it tries to find a compromise so that not only allows unimodal encoder to retain the heterogeneity of each modality, but also improve the compatibility between different modalities to enhance training efficiency and robustness.
> Therefore, it brings consistent performance improvement.
>
> [1] Yu, T., Kumar, S., Gupta, A., Levine, S., Hausman, K., & Finn, C. (2020). Gradient surgery for multi-task learning. Advances in neural information processing systems (NeurIPS), 33, 5824-5836.
>
> [2] Wei, Yake, and Di Hu. "Mmpareto: Boosting multimodal learning with innocent unimodal assistance." arXiv preprint arXiv:2405.17730 (2024).

---

### Official Review · Reviewer_2DRZ · 2025-11-01

**Soundness:** 3
**Presentation:** 3
**Contribution:** 2
**Rating:** 6
**Confidence:** 2

**Summary:**

This paper introduces GOAL (Gradient Orthogonalization and Adaptive Leveraging), a novel gradient modification method designed to address the modality imbalance problem in multimodal learning. GOAL combines two key components: Gradient Orthogonalization (GO), which resolves conflicting gradient directions across modalities, and Adaptive Leveraging (AL), which dynamically adjusts gradient magnitudes based on modality-specific uncertainties. Extensive experiments across various multimodal benchmarks demonstrate that GOAL consistently outperforms existing state-of-the-art methods in both efficiency and generalization. These results highlight GOAL's potential for improving multimodal optimization, providing a plug-and-play solution for a wide range of multimodal learning tasks.

**Strengths:**

- GOAL introduces an innovative method that combines Gradient Orthogonalization (GO) and Adaptive Leveraging (AL), effectively addressing the modality imbalance issue in multimodal learning. This method not only eliminates gradient direction conflicts across modalities but also adapts the gradient magnitude dynamically to handle modality uncertainty, demonstrating strong innovation.
- Through extensive experiments, the authors validate the effectiveness of GOAL on multiple multimodal benchmark datasets. The results show that GOAL significantly outperforms existing methods in efficiency and generalization, demonstrating its strong practicality and applicability. As a plug-and-play module, GOAL can be easily integrated into various multimodal learning frameworks, enhancing its versatility.
- The writing is clear, and the structure is well-organized, with the theoretical analysis and experimental sections supporting each other, allowing readers to easily understand the workings of GOAL and the experimental results. The presentation of the figures and tables is intuitive and effective, making it easy to understand the experimental process and the comparison of results.

**Weaknesses:**

- The GOAL method proposed in the paper is well-explained theoretically, but lacks specific guidance on how to adjust the method in practical applications. For example, how to tune hyperparameters and optimize the weighting strategy of the AL and GO components for different tasks and datasets is not thoroughly discussed.
- Although the paper demonstrates the superior performance of GOAL on multiple datasets, issues such as gradient conflict variations and dynamic weighting adaptability in more complex applications could affect the model's effectiveness. These potential challenges are not discussed in the paper.
- Although the ablation experiments show the individual effects of the AL and GO components, there is a lack of in-depth quantitative analysis on how they collaborate, interact across different datasets, and optimize in complex and imbalanced datasets.
- Although GOAL performs excellently, the paper does not sufficiently discuss its advantages and disadvantages in terms of computational cost, training time, and memory consumption, especially whether there are potential computational bottlenecks given that GOAL is a plug-and-play gradient modification method.
- Although the GOAL method performs excellently in enhancing multimodal learning, its internal mechanism, especially the interaction between GO and AL components, remains opaque. The lack of interpretability analysis may make it difficult for readers to understand how GOAL coordinates optimization across different modalities. Future work could provide deeper interpretability analysis to help readers better grasp how GOAL improves model performance.

**Questions:**

See weaknesses.

---

> ### Author Response · Authors · 2025-11-20
> **Part 1**
>
> **Weakness 1 :** The GOAL method proposed in the paper is well-explained theoretically, but lacks specific guidance on how to adjust the method in practical applications. For example, how to tune hyperparameters and optimize the weighting strategy of the AL and GO components for different tasks and datasets is not thoroughly discussed.
>
> **Answer:** We appreciate this suggestion. We would like to know which specific hyperparameters you are referring to:
>
> **Internal Module Hyperparameters:** If referring to the internal hyperparameters of the model modules, we wish to clarify that GOAL does not require specific hyperparameter tuning. The only value that might cause confusion is the temperature $T$ in Equation (7), which was fixed at 1 in all our experiments. However, we are happy to report a sensitivity analysis. The table below shows the impact of different $T$ values on performance across different datasets (also added to Appendix A.10).
>
>  **Table: Accuracy with different T values.**
>
>  | CREMA-D (T) | CREMA-D (Acc) | KS (T) | KS (Acc) | AVE (T) | AVE (Acc) |
>  | :---: | :---: | :---: | :---: | :---: | :---: |
>  | 0.2 | 76.08 | 0.2 | 74.35 | 0.2 | 70.15 |
>  | 0.4 | 77.96 | 0.4 | 74.49 | 0.4 | 71.23 |
>  | 0.6 | 77.42 | 0.6 | 74.63 | 0.6 | 70.23 |
>  | 0.8 | 77.82 | 0.8 | 74.91 | 0.8 | 71.98 |
>  | 1.0 | 78.23 | 1.0 | 75.37 | 1.0 | 72.14 |
>
> **Training Hyperparameters:** If referring to model training hyperparameters, these are detailed in Sections 4.1.1 and 4.1.2. We are happy to elaborate here: For video input, we adjust the sampling frames based on video length (2 frames per clip for CREMA-D, CMU-MOSI, MELD, and IEMOCAP; 3 frames for Kinetics-Sounds and AVE). We use the SGD optimizer with a momentum of 0.9 and weight decay of $1 \times 10^{-4}$. The learning rate is set to $4 \times 10^{-4}$ with a decay step size of 30. The batch size is 64. All experiments were conducted on an NVIDIA GeForce 5090 GPU.
>
> **Weakness 2 :** Although the paper demonstrates the superior performance of GOAL on multiple datasets, issues such as gradient conflict variations and dynamic weighting adaptability in more complex applications could affect the model's effectiveness. These potential challenges are not discussed in the paper.
>
> **Answer:** Indeed, your insight is profound and aligns perfectly with our thinking! We briefly mentioned this in the Conclusion/Future Work section of our paper. We agree that studying the behavior of GOAL in more complex or non-stationary environments is valuable. Current research on modal imbalance is mostly based on the "multi-encoder" architecture we mentioned, which is convenient for existing gradient manipulation methods. However, determining what new changes are needed for existing methods (including ours) when using more complex fusion modules is our direction for future research. As a preliminary step, we investigated tasks with more than two modalities and class imbalance (Table 3) and regression tasks (Table 4).
>
> **Weakness 3 :** Although the ablation experiments show the individual effects of the AL and GO components, there is a lack of in-depth quantitative analysis on how they collaborate, interact across different datasets, and optimize in complex and imbalanced datasets.
>
> **Answer:** Thank you for pointing out the need to better explain the synergy between the modules. The design of GOAL is **sequential**:
>
> Step 1 (AL): First, the AL module (Sec 3.3) addresses the "gradient magnitude inconsistency." Based on the "optimal imbalance" theory [1], it uses entropy as a variance estimator to adaptively scale gradients from the multimodal loss.
>
> Step 2 (GO): Then, the GO module (Sec 3.4) receives these AL-scaled gradients to address "gradient direction conflict." By projecting conflicting vectors, GO ensures the final gradient directions are synergistic. Our ablation study (Table 2) quantitatively demonstrates this synergy: The Baseline (69.62) improves with AL-only (75.13) and GO-only (76.75). However, GOAL (78.23) achieves performance higher than the sum of the individual gains of the two components, indicating a positive synergistic relationship between AL and GO.
>
> [1] Wei, Shicai, Chunbo Luo, and Yang Luo. "Improving multimodal learning via imbalanced learning." Proceedings of the IEEE/CVF International Conference on Computer Vision. 2025.

---

> > ### Author Response · Authors · 2025-11-20
> > **Part 2**
> >
> > **Weakness 4 :** Although GOAL performs excellently, the paper does not sufficiently discuss its advantages and disadvantages in terms of computational cost, training time, and memory consumption, especially whether there are potential computational bottlenecks given that GOAL is a plug-and-play gradient modification method.
> >
> >
> > **Answer:**
> > We thank the reviewer for this practical question. Compared to the network's forward/backward propagation, GOAL's computation is extremely lightweight. Theoretically, assuming $M$ modalities, batch size $B$, feature dimension $F$, and class number $C$:
> >
> > AL Complexity: $O(M \cdot B \cdot F \cdot C)$
> >
> > GO Complexity: $O(M \cdot B \cdot F)$
> >
> > Total Complexity: $O(M \cdot B \cdot F \cdot C + M \cdot B \cdot F)$.
> >
> > Specifically, taking the CREMA-D dataset with a ResNet18 backbone as an example, a single pass for bimodal (Audio+Visual) requires $\approx 3.6 \times 10^9$ FLOPs. In contrast, our GOAL module performs:AL step: $2 \times 32 \times 1024 \times 6 = 393,216$ operations.GO step: $2 \times 32 \times 1024 = 65,536$ operations.Total Extra: 458,752 operations, which is merely **0.0127%** of the backbone's computation.We have reported single-pass time, relative training time, and GPU memory usage for datasets of different scales in the new Table 8 (added to Appendix A.11).
> >
> > **Table: Computation time and memory consumption analysis per sample.**
> >
> >  | Metric | CREMA-D (Base) | CREMA-D (GOAL) | KS (Base) | KS (GOAL) | AVE (Base) | AVE (GOAL) |
> >  | :--- | :---: | :---: | :---: | :---: | :---: | :---: |
> >  | Avg time/sample (ms) | 15.67 | 17.41 | 19.68 | 20.47 | 21.72 | 23.28 |
> >  | Avg time/epoch (min) | 1.73 | 1.93 | 7.60 | 7.28 | 1.22 | 1.30 |
> >  | GPU usage (MiB) | 1089 | 1089 | 1093 | 1093 | 1093 | 1093 |
> >
> > **Weakness 5 :** Although the GOAL method performs excellently in enhancing multimodal learning, its internal mechanism, especially the interaction between GO and AL components, remains opaque. The lack of interpretability analysis may make it difficult for readers to understand how GOAL coordinates optimization across different modalities. Future work could provide deeper interpretability analysis to help readers better grasp how GOAL improves model performance.
> >
> > **Answer:** We agree that interpretability is crucial. Our current paper provides some insights: Figure 2b shows that GOAL achieves the highest feature correlation between modalities, implying better compatibility. In the long term, a promising direction is to explore feature-level interpretability: studying how fused representations change under GOAL. We will mention this as future work.

---

### Comment · Area_Chair_CYKx · 2025-11-24

Dear reviewers,

Thank you for your dedicated service as reviewers. Your efforts are critical to the success of our conference, and we deeply appreciate your time and expertise.

This paper has received reviews from reviewers but some have not provided a response to the author rebuttal. Given the limited time we have for author-reviewer discussions, we kindly ask you to share your post-rebuttal feedback to help clarify your perspective and aid the decision-making process.

Your input is invaluable in ensuring a fair and thorough review process.

Best,
AC

---

### Meta-Review · Area_Chair_sX9h · 2025-12-27

**Summary:**

This paper received initial ratings of 6, 4, 2 and 6 and corresponding confidence ratings of the reviewers are 2,4,5 and 3. The reviewers appreciated the presentation quality, evaluation on multiple datasets, and plug-and-play model. They raised questions and requested more clarifications, explanations and additional experiments. The authors replied to reviewers’ questions with additional clarifications, theoretical justification and experiments. Before the stop of the discussion period, only reviewer YvAS joined the discussion. The first authors’ response to reviewer YvAS does not fully address his concerns. He requested further justification, saying “lacks a strong justification”. The other three reviewers did not join the discussion before the stop of the discussion period. The AC reads their comments and authors’ responses. For reviewer 2DRZ’s weakness 2 and weakness 5, the authors did not fully answer the questions but said that “We briefly mentioned this in the Conclusion/Future Work section of our paper” and “We will mention this as future work.” (The AC noted that the authors provided some additional experiments and explanation for these two weaknesses.) The reviewers had different views on the novelty of this work. Reviewer YYZM commented on the novelty of this work and directly said that “The method seems incremental” but reviewer 2DRZ appreciated the innovative method and reviewer PpR3 considered that it is a novel multimodal optimization method. The AC reads the authors’ responses and a part of paper. He has the same view as the reviewer 2DRZ. For the novelty, the authors defenced their novelty by “identifying the new factor, i.e., the gradient direction in cross-modality compatibility”. This justification does not fully convince the AC since pervious work has discussed, e.g., cross-modality gradient realignment. According to the reviewers’ initial ratings, confidence, and authors’ responses, the AC does not recommend accepting this paper. The AC suggests the authors improve the work and consider another conference or journal.

**Reviewer Concerns:**

The authors did not fully answer reviewer 2DRZ’s weakness 2 and weakness 5.
The answer about novelty is not convinced.

**Reviewer Scores:**

I don't think that the reviewers will increase scores, in particular YYZM, whose initial score is 2.

---

### Decision · Program_Chairs · 2026-01-26

Reject